# Engineered droplet-forming peptide as photocontrollable phase modulator for fused in sarcoma protein

Hao-Yu Chuang [1,2,3], Ruei-Yu He [1], Yung-An Huang [1], Wan-Ting Hsu[1], Ya-Jen Cheng [4,5], Zheng-Rong Guo[6], Niaz Wali[1], Ing-Shouh Hwang [6], Jiun-Jie Shie [1] & Joseph Jen-Tse Huang [1,4,7,8] ✉

The assembly and disassembly of biomolecular condensates are crucial for the subcellular compartmentalization of biomolecules in the control of cellular reactions. Recently, a correlation has been discovered between the phase transition of condensates and their maturation (aggregation) process in diseases. Therefore, modulating the phase of condensates to unravel the roles of condensation has become a matter of interest. Here, we create a peptide-based phase modulator, JSF1, which forms droplets in the dark and transforms into amyloid-like fibrils upon photoinitiation, as evidenced by their distinctive nanomechanical and dynamic properties. JSF1 is found to effectively enhance the condensation of purified fused in sarcoma (FUS) protein and, upon light exposure, induce its fibrilization. We also use JSF1 to modulate the biophysical states of FUS condensates in live cells and elucidate the relationship between FUS phase transition and FUS proteinopathy, thereby shedding light on the effect of protein phase transition on cellular function and malfunction.

Biomolecular condensates formed through liquid–liquid phase separation (LLPS) play a pivotal role in biological systems and govern several vital cellular processes[1]. Through their assembly and disassembly, these condensates facilitate the subcellular compartmentalization of specific biomolecules, such as proteins and nucleic acids, and increase the local concentration of these biomolecules to regulate biological reactions[2]. Examples of well-known biomolecular condensates are nucleoli, which concentrate RNA polymerase I and ribosomal RNA for ribosome assembly[3], and centrosomes, which recruit tubulins and microtubule-associated proteins to accelerate microtubule formation during mitosis[4]. Biomolecular condensates also function as environmental sensors within cells, responding to changes in temperature, pH, and salt concentration[5]. Stress granules, for instance, form in the cytosol under conditions of stress but then swiftly disassemble once the stress has subsided. Stress granules halt cellular mRNA translation by rapidly assembling non-translating mRNAs and associated RNA-binding proteins[6]. Studies have suggested that specific chaperones, such as HspB8[7] and Hsp70[8], may be recruited into stress granules to maintain the fluidity of these granules and participate in the disassembly process.

Although the assembly of biomolecular condensates is often reversible, these condensates may also undergo irreversible phase transition. Recent studies have demonstrated that prolonged stress can impair the homeostasis of condensates and accelerate their "maturation". During the maturation process, the liquid condensates become more viscoelastic over time and eventually behave as solids[9–11]. The connection between these irreversible processes and neurodegenerative diseases has also come to light[12]. Notably, it has been shown that the maturation of biomolecular condensates could be promoted by mutations in DNA/RNA-binding proteins with low-complexity

[1]Institute of Chemistry, Academia Sinica, Taipei 115, Taiwan. [2]Chemical Biology and Molecular Biophysics, Taiwan International Graduate Program, Academia Sinica, Taipei 115, Taiwan. [3]Department of Chemistry, National Tsing Hua University, Hsinchu 300, Taiwan. [4]Neuroscience Program of Academia Sinica, Academia Sinica, Taipei 115, Taiwan. [5]Institute of Molecular Biology, Academia Sinica, Taipei 115, Taiwan. [6]Institute of Physics, Academia Sinica, Taipei 115, Taiwan. [7]Sustainable Chemical Science and Technology, Taiwan International Graduate Program, Academia Sinica, Taipei 115, Taiwan. [8]Department of Applied Chemistry, National Chiayi University, Chiayi City 600, Taiwan. ✉e-mail: jthuang@gate.sinica.edu.tw

domains (LCDs), including fused in sarcoma (FUS)[13], associated with amyotrophic lateral sclerosis (ALS)[14]. Therefore, understanding the molecular mechanism underlying the condensation and maturation of biomolecular condensates is crucial for delineating the physiology and pathology of various biological processes. Consequently, effective tools for modulating protein phases in live cells are required[15].

Among different DNA/RNA-binding proteins, FUS stands out due to its rapid recruitment to DNA damage sites[9], its ability to form droplets both in vivo[16] and in vitro[17], and its propensity for liquid-to-solid transitions, ultimately resulting in disease-related aggregates under pathological conditions[13,18]. Studies have emphasized the significance of tyrosine residues (denoted Y) in LCDs and arginine residues (denoted R) in arginine−glycine−glycine-rich (RGG) domains (Fig. 1a) in dominating its LLPS properties through cation−π and π−π interactions[18,19]. Furthermore, the antiparallel β-sheet motif in the LCD (FUS$_{39-95}$) was proven to play a key role in FUS self-assembly[20]. Additionally, short fragments of the FUS LCD (FUS$_{54-59}$ and FUS$_{50-65}$) that can form either reversible[21] or irreversible aggregates[22] have been observed.

Here, we create a peptide-based phase modulator capable of modulating FUS condensation and transforming FUS condensates into solid-like aggregates in response to external stimuli. Our peptide phase modulator, JSF1, is created by conjugating a FUS LCD fragment (FUS$_{50-60}$, YGQSSYSSYGQ; Fig. 1a) to a polyarginine tract (RRRRRR) through a photocleavable linker (methoxynitrobenzene; Fig. 1b). We demonstrate that JSF1 forms droplets under specific physiological conditions and undergoes a phase transition upon photoinitiation. In addition, we reveal that JSF1 promotes the LLPS of purified FUS under dark conditions and induces FUS aggregation under light exposure. Finally, we use JSF1 to "control" the biophysical states of FUS condensates in live cells, exploring the correlation between FUS phase transitions and FUS proteinopathy.

## Results

### The molecular design, preparation, and characterization of JSF1 condensates

The phase modulator, JSF1, was mainly composed of a FUS LCD fragment and a polyarginine tract (RRRRRR). By applying the protein aggregation predictor[23], we identified the protein segments (FUS$_{50-60}$: YGQSSYSSYGQ) in FUS LCD with high propensity for fibrilization (Supplementary Fig. 1). We surmised the multivalent interactions (e.g., cation−π and π−π interactions) between positively charged arginines and the three tyrosines in FUS$_{50-60}$ could benefit greatly on the droplet formation[18,19]. Meanwhile, the polyarginine tract could also provide cell

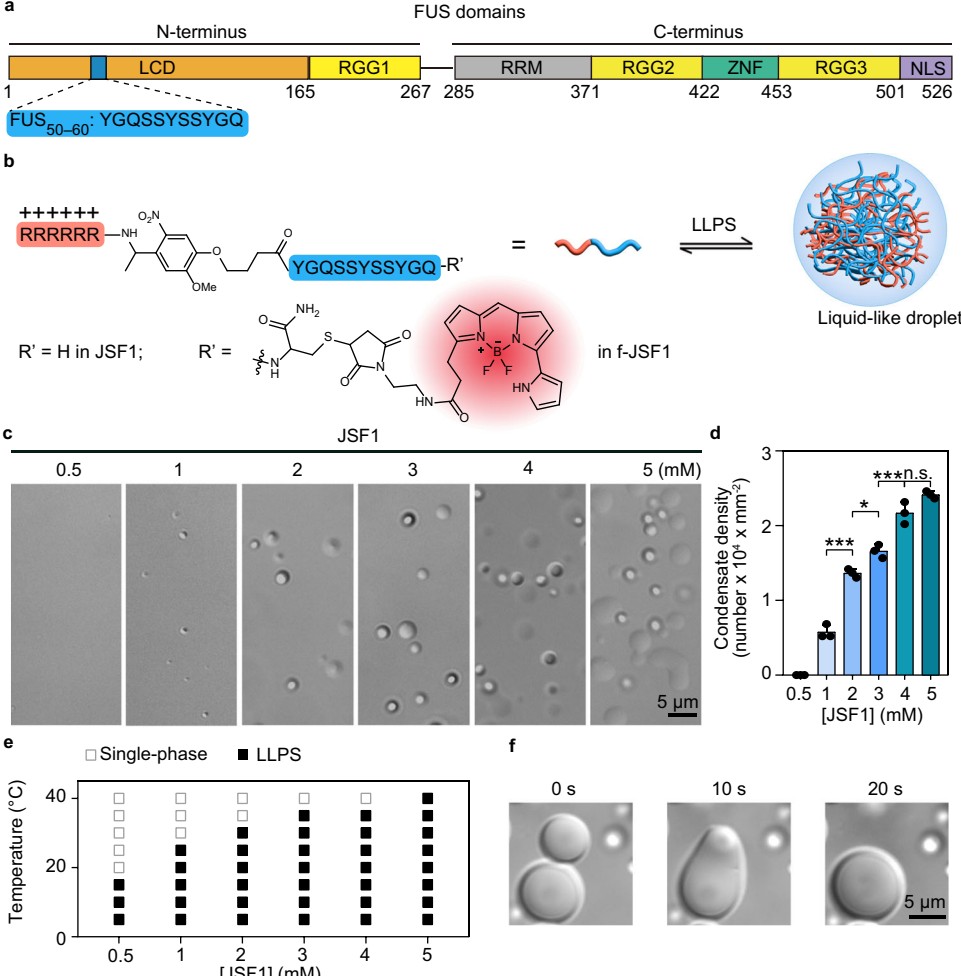

**Fig. 1 | The design and LLPS property of the droplet-forming peptide, JSF1.**
**a** Domains of FUS. **b** Schematic illustration of the design and LLPS of JSF1. f-JSF1: fluorophore-attached JSF1. **c** DIC images of 0.5 − 5 mM JSF1. **d** Condensate density of 0.5−5 mM JSF1. Condensates within an area = 0.006 mm² were analyzed. The statistic results were quantified by ImageJ and shown as mean ± SD of 3 independent replicates (n = 3). Data were analyzed by one-way ANOVA with Tukey post-hoc test with a 95% confidence interval. *P < 0.05, ***P < 0.001, n.s. non-significant. 1 mM vs 2 mM: P < 0.0001, q = 14.53, DF = 10. 2 mM vs 3 mM: P = 0.0218, q = 5.411, DF = 10. 3 mM vs 4 mM: P = 0.0004, q = 9.419, DF = 10. 4 mM vs 5 mM: P = 0.0586, q = 4.509, DF = 10. **e** Phase diagram of JSF1 as the function of JSF1 concentration and temperature. **f** The fusion of JSF1 (3 mM) condensates in the presence of PEG$_{8000}$ (30%). Source data are provided as a Source Data file.

penetrating ability[24]. To further enable the photocontrollable ability of this phase modulator, the photocleavable linker was applied to conjugate FUS LCD fragment with the polyarginine tract.

JSF1 was synthesized through solid-phase peptide synthesis, characterized using reverse-phase high-performance liquid chromatography (Supplementary Fig. 2a), and confirmed through matrix-assisted laser desorption/ionization mass spectrometry (Supplementary Fig. 2b). Differential interference contrast (DIC) microscopy was employed to monitor the room-temperature LLPS of JSF1 at various concentrations in phosphate buffer (100 mM $K_2HPO_4/KH_2PO_4$, pH 7.0; details in the Methods section). Our results showed that spherical JSF1 droplets formed when the JSF1 concentration was 1 mM or higher (Fig. 1c). The density and diameter of these droplets were positively correlated with the JSF1 concentration (Fig. 1d and Supplementary Fig. 3). By contrast, neither polyarginine tract nor FUS$_{50-60}$ could form droplets (Supplementary Fig. 2c–f and Fig. 4). To map the phase diagram of JSF1 in terms of its concentration and the temperature, we measured the turbidity of JSF1 at various temperatures. The results revealed that decreasing the temperature from 40 to 5 °C considerably increased the turbidity of JSF1 solutions of all JSF1 concentrations (0.5–5 mM; Supplementary Fig. 5a), indicating a negative correlation between turbidity and temperature. During the measurement, we also noticed that the LLPS of JSF1 could be reversed through temperature cycling (Supplementary Fig. 5b). By calculating the critical temperature of JSF1 at various concentrations (0.5–5 mM), we obtained a phase diagram that illustrates the single-phase state and LLPS state (Fig. 1e; details in the Methods section). Since JSF1 solution underwent LLPS at lower temperature and was miscible at higher temperature, the process followed the upper critical solution temperature (UCST) behavior[25]. Additionally, we confirmed that the LLPS of JSF1 was suppressed (Supplementary Fig. 6) at higher KCl concentration, suggesting electrostatic interactions such as cation–π and dipole–dipole interactions were reduced at stronger ionic strength[26]. It is worth to note that JSF1 condensation could be enhanced by the crowding agent PEG$_{8000}$ (0 – 30%, Supplementary Fig. 7) and the fusion events could be clearly observed (Fig. 1f and Supplementary Video 1).

## JSF1 condensates underwent liquid-to-solid phase transition upon photoinitiation and eventually formed amyloid-like fibrils

Because JSF1 could form condensates, we next determined whether irradiation could lead to release of the polyarginine tract, induce the phase transition of JSF1, and result in the development of mature fibrils composed of FUS$_{50-60}$ (Fig. 2a). JSF1 condensates (3 mM) were irradiated (light of wavelength 365 nm and power density 165 mW/cm²) for

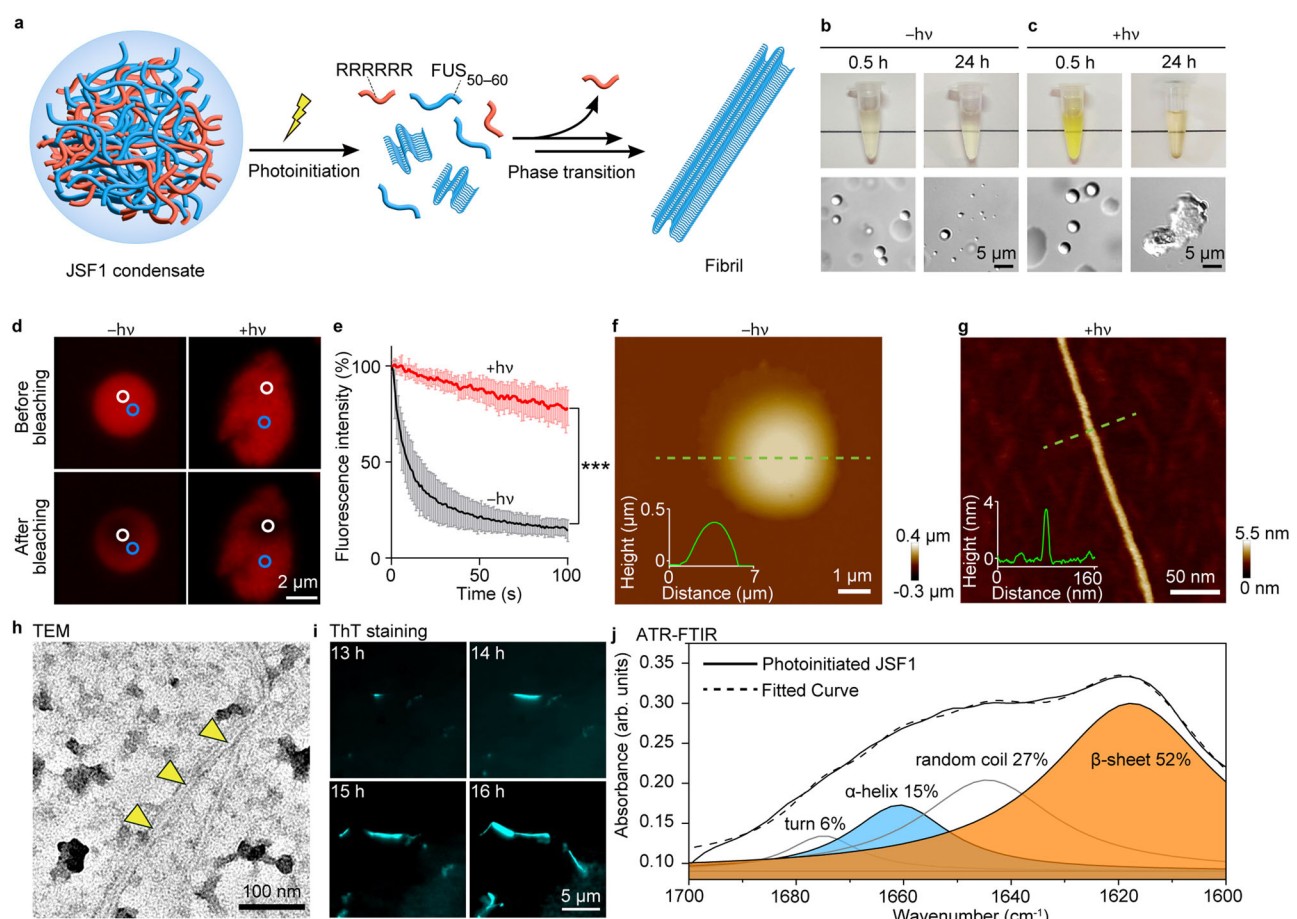

Fig. 2 | JSF1 condensates underwent liquid-to-solid phase transition to form amyloid-like fibrils after photoinitiation. a Schematic illustration of JSF1 phase transition after photoinitiation. b Photographs and DIC images of JSF1 without photoinitiation incubated for 0.5 and 24 h. c Photographs and DIC images of JSF1 with photoinitiation incubated for 0.5 and 24 h. d FLIP representative images of JSF1 condensate (−hν) and aggregate (+hν). White circle: bleached zone. Blue circle: region of interest (ROI). e FLIP traces of JSF1 condensates (−hν) and aggregates (+hν). The statistic results were shown as mean ± SD (n = 5). Data were analyzed by two-way ANOVA using Sidak post-hoc test with a 95% confidence interval. At the time point = 100 s, ***P < 0.001. + hν vs −hν: P < 0.0001, t = 13.53, DF = 909. f, g AFM height image of (f) JSF1 condensate (−hν) and (g) fibril (+hν). Height profiles along the green line were shown in the section. The experiment was repeated for 3 biological replicates. h TEM image of photoinitiated JSF1. The experiment was repeated for 3 biological replicates. Yellow arrow: the fibril. [JSF1] = 3 mM and incubation time was 24 h. i TIRF images of 100 μM photoinitiated JSF1 with ThT and incubated for 13 – 16 h. j ATR-FTIR and deconvolution results of 3 mM photoinitiated JSF1. Source data are provided as a Source Data file.

3 min, and we observed consequent photocleavage and release of FUS$_{50-60}$ and the polyarginine tract, as confirmed through ultraviolet–visible (UV-Vis) spectroscopy, reverse-phase high-performance liquid chromatography, and matrix-assisted laser desorption/ionization mass spectrometry (Supplementary Fig. 8). The JSF1 solution (3 mM) was cloudy and contained spherical condensates under differential interference contrast microscopy (Fig. 2b). After 24 h of incubation, these condensates settled down to the bottom of the tube and make the solution transparent (Fig. 2b). Upon photocleavage, the solution turned yellow at 0.5 h and the spherical morphology of these condensates persisted (Fig. 2c). After further incubation, brown depositions were found at the bottom of the tube and irregular aggregates were revealed (Fig. 2c) at 24 h. Additionally, time-course DIC (Supplementary Fig. 9) demonstrated that the photoinitiated condensates became non-spherical after 14 h of incubation and gradually transformed into aggregates. Note that polyarginine tract did not form aggregates after 24 h of incubation (Supplementary Fig. 10). On the contrary, FUS$_{50-60}$ formed huge aggregates in buffer (Supplementary Fig. 4), indicating the aggregates of photoinitiated JSF1 were formed by FUS$_{50-60}$.

To further investigate the biophysical properties of JSF1 condensates and aggregates, we employed the fluorescence loss in photobleaching (FLIP) assay to monitor changes in fluidity[9,27]. In this assay, a bleached zone was repetitively photobleached, while the fluorescence intensity decay of a region of interest (ROI) different from the bleached zone was monitored. In this study, we created fluorophore-attached JSF1 (denoted f-JSF1, Fig. 1b, details in the Supplementary Method and in Supplementary Figs. 11, 12a, b) and spiked the JSF1 condensates with it (f-JSF1:JSF1 ratio = 1:2500; Supplementary Fig. 12c). After 24 h of incubation, the samples with or without photoinitiation were applied to FLIP assay. The fluorescence intensity of the condensates without photoinitiation was discovered to rapidly decrease (to 15% at $t = 100$ s; Fig. 2d left panels and Fig. 2e), indicating the high fluidity of the condensates. By contrast, the fluorescence intensity of the aggregates formed after photoinitiation decreased slowly (to 78% at $t = 100$ s; Fig. 2d right panels and Fig. 2e), suggesting much lower fluidity of these aggregates. In summary, with the aid of photoinitiation, JSF1 condensates could be transformed into solid aggregates with significantly lower fluidity.

Differences in nanomechanical properties were also monitored to examine the phase transition between condensates and aggregates. We used the PeakForce Quantitative Nano-Mechanics mode of atomic force microscopy (AFM) to simultaneously perform topographic imaging and mapping of nanomechanical properties. In the absence of photoinitiation, JSF1 formed a dome-shaped structure with a lateral diameter of approximately 5 μm and a maximum height of approximately 400 nm on a flat substrate (Fig. 2f). After photoinitiation, straight fibrils were observed (Fig. 2g) that had a width of approximately 18 nm and a maximum height of approximately 4 nm. The apparent width was larger than the real width due to the broadening effect of the finite size of the atomic force microscopy probe. The mapping of Young's modulus revealed darker contrast (lower stiffness) for the condensates and fibrils than for the substrate, indicating their soft nature (Supplementary Fig. 13a and b). The average Young's modulus of the fibrils (439 ± 83 MPa) was roughly five times that of the condensates (87 ± 22 MPa; Supplementary Fig. 13c), suggesting that the JSF1 nanofibrils were significantly stiffer. The condensates exhibited a low stiffness (Fig. 13a and 13c) and a smooth surface morphology (Fig. 2f), which indicate that they were in a liquid-like state. The detailed morphology of the JSF1 nanofibrils was also confirmed using transmission electron microscopy (TEM) which revealed a comparable width of approximately 5 nm (Fig. 2h and Supplementary Fig. 14). The fibrilization processes of photoinitiated JSF1 were recorded using total internal reflection fluorescence (TIRF) microscopy and thioflavin T (ThT) staining (Video 2, Fig. 2i, Supplementary Fig. 15). Additionally, the β-sheet content of JSF1 before and after photoinitiation was revealed through deconvolution of attenuated total reflectance Fourier-transform infrared (ATR-FTIR) spectra (Fig. 2j and Supplementary Fig. 16). We conclusively demonstrated that JSF1 condensates underwent the liquid-to-solid phase transition upon photoinitiation and eventually formed amyloid-like fibrils.

## JSF1 served as a phase modulator for FUS, promoting LLPS under dark conditions and triggering FUS fibrilization upon photoinitiation

Once we had confirmed the dual biophysical states (i.e., condensates and fibrils) of JSF1 before and after photoinitiation, we investigated whether JSF1 could serve as a phase modulator against FUS condensates. Small molecules (e.g., bis-ANS and Congo red[15]) and some biomacromolecules (e.g., RNA and chaperones[8,28]) have been shown to act as phase modulators, promoting or inhibiting condensate formation through multivalent interactions[29]. To obtain further evidence of this role, a recombinant protein, His$_6$-MBP-(TEV recognition site)-FUS, was expressed in E. coli and purified by following the published protocol (Supplementary Fig. 17a and b; details in the Methods section)[30]. After the His$_6$-MBP tag was removed from the recombinant protein, FUS formed spherical condensates at a concentration equal or higher than 1.25 μM (Fig. 3a and Supplementary Fig. 17c). To determine whether JSF1 could modulate the LLPS of FUS, we incubated FUS condensates (2.5 μM) with various concentrations of JSF1 (6.25–100 μM). Note that JSF1 alone did not form condensates at these concentrations (Supplementary Fig. 18). Our results showed that the density of FUS condensates was increased by JSF1 (Fig. 3b and c) and the critical concentration of FUS LLPS could be reduced by JSF1 (Supplementary Fig. 19), suggesting JSF1 could facilitate FUS LLPS. Interestingly, polyarginine tract could also reduce the critical concentration of FUS (Supplementary Fig. 19), confirming the importance of arginines in providing the multivalency required for LLPS. By incubating Alexa Fluor-488-labeled FUS (denoted FUS-488) with f-JSF1 mixture (f-JSF1:JSF1 = 1:2500), we confirmed that f-JSF1 colocalized with FUS condensates (Supplementary Fig. 20). A fluorescence loss in photobleaching assay (Fig. 3d and e) further confirmed the high fluidity of FUS-488 and f-JSF1 in the mixture, suggesting that JSF1 underwent co-phase separation with FUS[26,27].

Because photoinitiated JSF1 could form amyloid-like fibrils, we examined whether it could also "seed" the formation of fibrils from FUS protein in vitro[22,31]. We incubated FUS condensates (2.5 μM) in the absence or presence of JSF1 and under or not under irradiation and monitored their morphology through differential interference contrast microscopy (details in the Methods section). After 48 h, we observed no change in the spherical morphology of condensates without JSF1 or without photoinitiation (−JSF1−hν, −JSF1+hν, and +JSF1−hν panels in Fig. 3f; condensates indicated by blue arrows). By contrast, when condensates were incubated with photoinitiated JSF1, fibril-like aggregates were observed (+JSF1+hν panel in Fig. 3f, aggregates indicated by yellow arrows). Together with the condensate density analysis (Fig. 3g), our data suggested that photoinitiated JSF1 triggered the transition of FUS from the condensate phase to the solid aggregate phase. Note that polyarginine tract failed to trigger the liquid-to-solid phase transition of FUS (Supplementary Fig. 21), indirectly reflecting the importance of FUS$_{50-60}$. Additionally, we performed transmission electron microscopy and immunogold labeling with FUS antibody to confirm that photoinitiated JSF1 promoted the formation of FUS-specific fibrils (Fig. 3h and Supplementary Fig. 22). JSF1 was thus discovered to serve as a phase modulator in vitro to co-phase separate with FUS under dark conditions and trigger the transition of FUS condensates into fibrils under irradiation conditions (Fig. 3i).

## JSF1 phase modulator regulated the fluidity of FUS condensates in cells and further influenced cytotoxicity

After confirming that JSF1 could serve as a phase modulator for FUS, we investigated whether JSF1 could also regulate the biophysical states

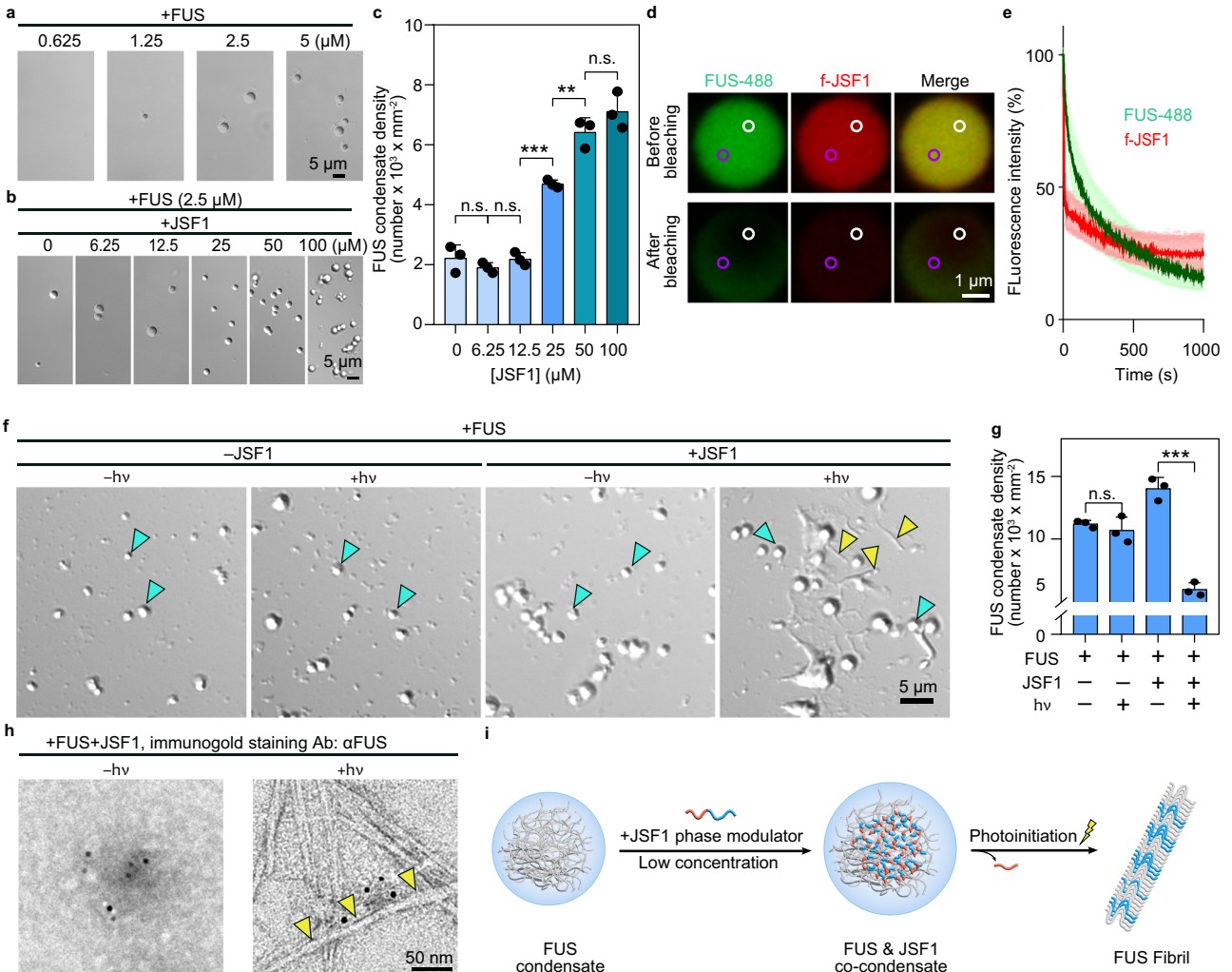

**Fig. 3 | JSF1 served as a phase modulator for FUS, promoting LLPS under dark conditions and triggering fibrilization upon photoinitiation. a** DIC images of FUS (0.625 – 5 μM). **b** DIC images of FUS (2.5 μM) with JSF1 (0 – 100 μM). **c** Condensate density of FUS (2.5 μM) with JSF1 (0–100 μM). Condensates within 0.01 mm² were analyzed. The statistic results were quantified by ImageJ and shown as mean ± SD of 3 independent replicates (*n* = 3). Data were analyzed by one-way ANOVA using Tukey post-hoc test with a 95% confidence interval. **P < 0.01, ***P < 0.001, n.s. non-significant. 0 μM vs 6.25 μM: *P* = 0.9074, q = 1.422, DF = 12. 6.25 μM vs 12.5 μM: *P* = 0.9354, q = 1.293, DF = 12. 12.5 μM vs 25 μM:*P* < 0.0001, q = 11.25, DF = 12. 25 μM vs 50 μM: *P* = 0.0015, q = 7.757, DF = 12. 50 μM vs 100 μM: *P* = 0.3073, q = 3.103, DF = 12. **d** FLIP representative images of FUS (2.5 μM, FUS-488:FUS = 1:9) and JSF1 (2.5 μM, f-JSF1:JSF1 = 1:19) co-condensate. White circle: bleached zone. Purple circle: region of interest (ROI). **e** FLIP traces of the co-condensates. Data were shown as mean ± SD (*n* = 5). **f** DIC images of FUS incubated in the presence or absence of JSF1 with or without photoinitiation for 48 h. Blue arrow: condensate. Yellow arrow: fibril-like aggregate. **g** Condensate density of the samples in (**f**). The statistic results were quantified by ImageJ and shown as mean ± SD of 3 independent replicates (*n* = 3). Data were analyzed by one-way ANOVA using Tukey post-hoc test with a 95% confidence interval. ***P < 0.001, n.s. non-significant. −JSF1−hν vs −JSF1+hν: *P* = 0.8330, q = 1.192, DF = 8. −JSF1−hν vs −JSF1+hν: *P* < 0.0001, q = 18.54, DF = 8. **h** TEM images with immunogold labeling of FUS incubated with JSF1 with or without photoinitiation at 48 hours. Antibody: αFUS. Yellow arrow: fibril. **i** Schematic illustration of the impacts of low concentration JSF1 on FUS condensates. Source data are provided as a Source Data file.

of FUS in the neuroblastoma 2 A (N2A) cell line. The N2A cell line was selected as the model because it is commonly used in studies related to neuronal function[32,33]. We added f-JSF1 (10 μM) to N2A cells and found that f-JSF1 successfully penetrated the N2A cells and was mainly localized in the cytosol (Fig. 4a). Notably, JSF1 did not have a toxic effect on the cells (Supplementary Fig. 23a). Because wild-type FUS is mainly localized in the nuclear region of cells[18], the possible interaction between JSF1 and wild-type FUS was limited. To further explore the potential effect of JSF1 on FUS, we used mutated FUS (FUS^R522G) as a model system because it forms cytosolic condensates when overexpressed in cells and transforms into aggregated structures under pathological conditions[16,34,35].

N2A cells overexpressing FUS^R522G–enhanced green fluorescent protein (EGFP) were incubated with f-JSF1 (10 μM) for 24 h. FUS^R522G–EGFP readily formed cytosolic condensates which colocalized with f-JSF1 (Fig. 4b and c) but not fluorophore-attached polyarginine tract (Supplementary Fig. 24). The cells were immunostained with G3BP1 antibody, and this revealed that the FUS^R522G–EGFP condensates were recruited into stress granules (Supplementary Fig. 25). We performed a fluorescence recovery after photobleaching (FRAP) assay to analyze the fluidity of the FUS^R522G–EGFP condensates in live cells in the presence or absence of JSF1 under or not under photoinitiation (details in the Methods section). Our results suggested that the FUS^R522G–EGFP condensates retained high fluidity in the absence of JSF1, given that the fluorescence intensity rapidly recovered to 60% within 40 s (−JSF1−hν and −JSF1+hν panels in Fig. 4d, e). By contrast, in the presence of photoinitiated JSF1, the fluidity of the FUS^R522G–EGFP condensates significantly decreased ( +JSF1+hν panels in Fig. 4d and e), indicating that photoinitiated JSF1 induced the transformation of FUS^R522G–EGFP condensates into aggregated structures. In fact, through western

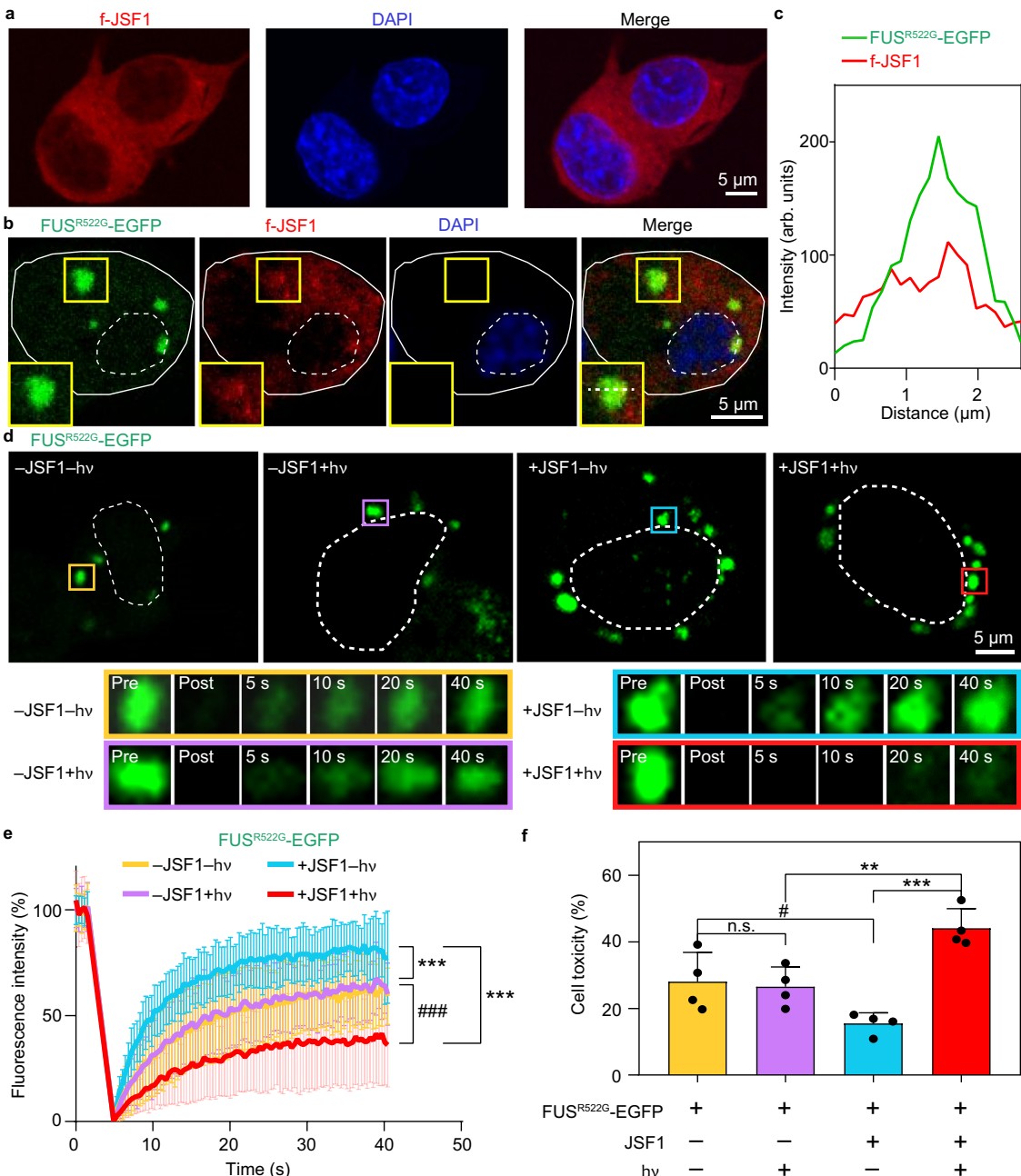

**Fig. 4 | JSF1 modulated the biophysical states of FUS^R522G-EGFP and cytotoxicity in N2A cells. a** JSF1 penetrated into N2A cells and localized in the cytosol. **b** f-JSF1 colocalized with cytosolic FUS^R522G-EGFP condensates. Solid line: cell. Dashed line: nucleus. Section: colocalized condensate. The section profile along the dashed line was shown in (**c**). **d** Representative images of FUS^R522G-EGFP FRAP assay. Box: bleached condensate. The enlarged bleached condensates at the time points (Pre: pre-bleach, Post: post-bleach) were shown below. Dashed line: nucleus. **e** FRAP traces of FUS^R522G–EGFP condensates mentioned in (**d**). The statistic results were shown as mean ± SD ($n \geq 40$). Data were analyzed by two-way ANOVA using Tukey post-hoc test with a 95% confidence interval. At the time point = 40 s, *** or ###

$P < 0.001$. –JSF1–hν vs +JSF1–hν: $P < 0.0001$, q = 6.469, DF = 13400. +JSF1–hν vs +JSF1+hν: $P < 0.0001$ q = 15.41, DF = 13400. –JSF1+hν vs +JSF1+hν: $P < 0.0001$, q = 8.91, DF = 13400. **f** The cell toxicity in the JSF1-treated or control cells in the presence and absence of photoinitiation. The statistic results were shown as mean ± SD of 4 independent replicates ($n = 4$). Data were analyzed by one-way ANOVA using Tukey post-hoc test with a 95% confidence interval. #$P < 0.05$, **$P < 0.01$, ***$P < 0.001$, n.s. non-significant. –JSF1–hν vs –JSF1+hν: $P = 0.9980$, q = 0.5977, DF = 18. –JSF1–hν vs +JSF1–hν: $P = 0.0272$, q = 4.922, DF = 18. –JSF1+hν vs +JSF1+hν: $P = 0.0015$, q = 6.886, DF = 18. +JSF1–hν vs +JSF1+hν: $P < 0.0001$, q = 11.21, DF = 18. Source data are provided as a Source Data file.

blotting of the RIPA-soluble and insoluble fractions of cell lysates, we confirmed that photoinitiated JSF1 led to higher concentrations of insoluble FUS^R522G–EGFP (Supplementary Fig. 26, details in the Methods section). Interestingly, we also found that JSF1 increased the fluidity of the FUS^R522G–EGFP condensates under dark conditions, given that fluorescence intensity recovered to 80% during the fluorescence recovery after photobleaching analysis ( + JSF1–hν panels in Fig. 4d and e). Based

on the FRAP analysis and western blotting, we showed that only the photoinitiated JSF1 could trigger the reduction of FUS fluidity and increase the aggregation of FUS.

Because the aggregation of FUS in stress granules is strongly correlated with FUS proteinopathy[36], we further examined the effect of FUS^R522G–EGFP modulation on cell viability. The cells incubated with photoinitiated JSF1 ( + JSF1+hν) had much greater toxicity than did those

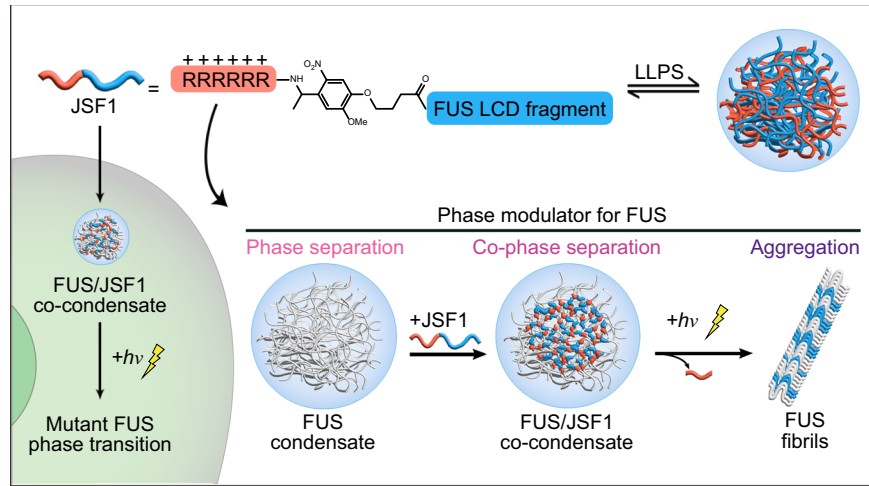

**Fig. 5 | Schematic illustration of JSF1 as a dual-function phase modulator for FUS protein.** JSF1 undergoes LLPS to form droplets. When added to FUS condensates, JSF1 enhances its LLPS in the dark and further triggers its fibrilization upon photoinitiation. JSF1 can also modulate the phases of mutant FUS condensates in live cells.

without JSF1 or photoinitiation ($-$JSF1$-$hv, $-$JSF1+hv, and +JSF1$-$hv), suggesting that the JSF1-induced aggregation of FUS[R522G]$-$EGFP was toxic (Fig. 4f). The cytotoxic effect was weaker when the JSF1 was not photoinitiated (+JSF1$-$hv). Meanwhile, all control experiments with the non-transfected conditions exhibited negligible toxicity (Supplementary Fig. 23b). Our data suggests that maintaining the fluidity of cytosolic FUS condensates in stress granules may be beneficial to the cells. Taken together, the engineered condensate-forming peptide JSF1 was found to penetrate cells and colocalize with stress granules. JSF1 also functioned as a phase modulator to increase the fluidity of cellular FUS condensates under dark conditions, resulting in higher cell viability. By contrast, photoinitiated JSF1 triggered the transformation of cytosolic FUS condensates into toxic aggregates.

## Discussion

As recent literature has disclosed a potential correlation between the dysregulated LLPS and various diseases, extensive efforts have been made in developing different methods for regulating protein phase transitions within cells[14]. Some approaches have focused on modulating the phase separation of RNA-binding proteins using the Cry2-based optogenetic platform[37], enzymatically-triggered system[38], or small naphthalene sulfonate derivatives like bis-ANS and Congo Red[39]. Others have induced the aggregation of cytosolic proteins by peptides[40–42] or chemical chaperones such as glycerol and trehalose[43]. In contrast to these strategies, our phase modulator JSF1 serves as a dual-function phase modulator: it enhances protein phase separation in the absence of light and induces protein aggregation in the presence of light. With this versatile tool at our disposal, we can now delve deeper into the specific roles of phase transitions in the pathophysiology of various diseases. The maturation of RNA-binding proteins in stress granules has been shown to be related to neurotoxicity[36,39,44]. Consistent with these findings, our data indicate that decreasing the fluidity of FUS within stress granules increases cytotoxicity in N2A cells (Fig. 4d–f). This discovery confirms the significance of protein phase transitions in disease and sheds light on the development of novel therapeutic strategies based on protein phase modulation for treating neurodegenerative disorders.

LLPS is initiated by a nucleation process and is followed by growth[45]. The key component driving nucleation is called a scaffold, and other recruited molecules that form co-condensates with the scaffold are termed clients[29,46]. The specific interactions between clients and scaffolds govern the recruitment of clients into condensates[47–49]. Moreover, clients can regulate the stability of scaffold condensates. In brief, low-valency clients may destabilize condensates by competing for binding sites on scaffolds, whereas high-valency clients can provide additional crosslinking between scaffolds to either stabilize or promote their LLPS[50]. In our in vitro experiment, we observed that FUS formed condensates when the FUS concentration was 2.5 µM, and addition of JSF1 increased the density of these FUS condensates (Fig. 3b and c). Accordingly, FUS was considered as a scaffold accommodating its client, JSF1, which exhibited high valency, thereby offering extra crosslinking between FUS scaffolds. We also deduced that this high valency enhanced the fluidity of FUS condensates (Fig. 4d and e).

Recent studies have elucidated the driving force behind FUS condensation. Through the sequence analysis[2], truncation studies[30], and FUS mutation analysis[18], the LLPS of FUS has been shown to be primarily driven by the LCD and RGG domains of FUS (Fig. 1a) with cation–π and π–π interactions[19]. We surmised that the FUS LCD fragment (FUS[50–60]) and the cationic polyarginine tract significantly contribute to these multivalent interactions which are essential for LLPS. In addition, through the use of various bioinformatics tools (e.g., PONDR[51], PLAAC[52], PScore[53], and PASTA 2.0[23]), FUS[50–60] was a highly disordered sequence (Supplementary Fig. 27a) that has prion-like properties (Supplementary Fig. 27b), strong π–π interactions (Supplementary Fig. 27c), and a high tendency to from aggregate (Supplementary Fig. 1). These features likely account for the amyloid-like properties (Fig. 2) and seeding capacity (Fig. 3f–h) of the photocleaved FUS[50–60] peptide from JSF1.

In conclusion, we have created a droplet-forming peptide, JSF1, which can undergo LLPS under dark conditions and transform into amyloid-like fibrils upon photoinitiation. The biophysical and nanomechanical properties of these JSF1 condensates and fibrils were characterized and compared in this study. Leveraging its dual functionality, JSF1 can serve as a versatile phase modulator in vitro to co-phase separate with FUS and augment FUS LLPS (Fig. 5). Following photocleavage, photoinitiated JSF1 effectively seeds the formation of fibrils from FUS condensates. Cellular studies revealed that the fluidity of FUS in stress granules is increased by JSF1 but decreased by photoinitiated JSF1 (Fig. 5). We further confirmed that the fluidity of FUS is positively correlated with cell viability. Our findings offer a promising approach for modulating the biophysical states of proteins in the cellular environment and can thus benefit explorations of the function and malfunction of protein condensates as well as the consequent implications regarding the pathogenesis of neurodegenerative diseases.

## Methods

### Peptide synthesis

Both amino acids and 4-{4-[1-(9-fluorenylmethyloxycarbonylamino) ethyl]-2-methoxy-5-nitrophenoxy}-n-butanoic acid (Fmoc-photolabile

linker) were purchased from Advanced ChemTech. Peptides were synthesized by the standard Fmoc polyamide chemistry on Rink amide (RAM) resin using the Liberty Blue automated microwave peptide synthesizer (CEM, USA). After cleavage from the resin, purity of peptides was verified by high-performance liquid chromatography (HPLC) (1260 Infinity LC system, Agilent, USA) equipped with a C18 reversed-phase semipreparative column (Shiseido, Japan). The gradient separation was achieved by mixing buffer A (5% acetonitrile/0.1% TFA/94.9% water) and buffer B (0.1% TFA/99.9% acetonitrile). The flow rate was kept at 3 mL/min. The molecular weights of the peptides were identified by matrix-assisted laser desorption/ionization (MALDI) (Applied Biosystem, USA) mass spectrometry.

## LLPS sample preparation
Peptides were weighted and dissolved in ddH₂O to make a 10 mM stock solution followed by 1 min sonication. JSF1 stock solution was mixed with 1 M $K_2HPO_4/KH_2PO_4$ buffer (pH 7.0), 2 M KCl, and 60% polyethylene glycol 8000 ($PEG_{8000}$) to reach the desired working concentration (100 mM $K_2HPO_4/KH_2PO_4$ with desired KCl and $PEG_{8000}$ concentration). The samples were vortexed for 30 seconds and sonicated for 1 minute to ensure that everything was dissolved and mixed well.

## Differential interference contrast (DIC) microscopy
DIC imaging was carried out using an automated Nikon TiE microscope. An aliquot of 200 μL was transferred from Eppendorf to a 35 mm glass-bottom dish (Ibidi, Germany) before imaging. Images and videos were collected using a 60X oil immersion objective and recorded by an Andor iXon3 888 EMCCD camera.

## Condensate density and diameter quantification
DIC images of condensates were taken after 10 min of incubation after samples mounted on the glass-bottom dish and further analyzed by ImageJ. The condensates were manually counted, and the diameters were measured. For condensate densities, JSF1 condensates within 0.06 mm² or FUS condensates within 0.01 mm² were analyzed. For condensate diameter, more than 50 condensates were analyzed for each condition.

## Turbidity assay and phase diagram
Turbidity was measured by the optical density at 600 nm (O.D. 600 nm). Sample solutions were loaded in a 1 mm quartz cell and recorded by a J-815 CD spectrometer (JASCO, Japan). For temperature-dependent measurements, the starting temperature was 5 °C and raised with a gradient of 1 °C/min. The O.D. 600 nm was recorded every 5 °C. The phase diagram in Fig. 1e was derived from Supplementary Fig. 5a. The critical temperature was estimated by calculating the x-intercept of the tangent at the inflection point of the curve.

## Photoinitiation
JSF1 (3 mM) in 100 mM $K_2HPO_4/KH_2PO_4$ was photolinitiated by a UV LED Spot Curing System (UVATA, China). A 365 nm 165 mW/cm² UV light was applied to the sample for 30 seconds, and there was a 1-minute break before the next exposure to cool down the sample temperature. Six cycles were executed for a total of 3 min of exposure. To monitor the cleavage, samples were diluted 10-fold with 50% acetonitile and analyzed by HPLC and MALDI mass spectrometry.

## f-JSF1 and f-RRRRRR labeling
The fluorophore 3-((2Z)-2-{[1-(difluoroboryl)-5-1H-pyrrol-2-yl-1H-pyrrol-2-yl]methylene}-2H-pyrrol-5-yl)-N-[2-(2,5-dioxo-2,5-dihydro-1H-pyrrol-1-yl)ethyl]propenamide (JJS-0341; detailed synthesis procedures are described in Supplementary Method Section) was freshly dissolved in DMSO to make a 5 mM stock solution. JSF1 conjugated to a C-terminal cysteine (named JSF1C) was dissolved in 100 mM $K_2HPO_4$/

$KH_2PO_4$ buffer (pH 8.0) to make a 500 μM stock. JSF1C and the fluorophore stock solutions were mixed to prepare a solution containing 250 μM JSF1C and 250 μM fluorophore. The mixture was incubated at 37 °C with 750 rpm shaking for 1 h. For the labeling of polyarginine tract, Rhodamine B was conjugated to the N-terminal through piperazine and succinic anhydride before peptide cleavage from resin (denoted as f-RRRRRR). f-JSF1 and f-RRRRRR were purified by HPLC and identified by MALDI mass spectrometry as described in the "Peptide Synthesis" section.

## Fluorescence loss in photobleaching (FLIP)
For JSF1 condensates and aggregates, 1.2 μM f-JSF1 was spiked into 3 mM JSF1 (f-JSF1:JSF1 = 1:2500) in 100 mM $K_2HPO_4/KH_2PO_4$ with or without photoinitiation. After incubation for 24 h at 20 °C, the samples were analyzed by FLIP. For FUS and JSF1 co-condensates, 2.5 μM FUS solution (FUS-488:FUS = 1:9) and 25 μM JSF1 solution (f-JSF1:JSF1 = 1:19) were prepared in the buffer (100 mM Tris/HCl, pH 7.0, 150 mM KCl) and analyzed immediately after being freshly prepared. To conduct FLIP, samples were loaded on a POC-R2 cell cultivation chamber (PeCon) with the FoilCover Set to prevent evaporation. FLIP experiments were performed by defining regions of interest (ROIs), including the bleached zone ($ROI_b$), the non-bleached ROI ($ROI_{nb}$), and the background ROI ($ROI_{bg}$). A circular area of $ROI_b$ with a radius of approximately 1 μm was bleached in the condensate or aggregate with a radius between 3 μm and 10 μm. The $ROI_b$ were repeatedly bleached after ten acquisitions with reduced laser power (0.2% output) at the start of the experiment and after each bleach. A 561 nm laser was used at 100% by 20 iterations for the bleaching pulses for the JSF1 sample. A 480 nm lasers were used at 100% by 20 iterations for the bleaching pulses for the FUS and JSF1 co-condensate sample. An eventual pause between the bleaches ensured no recovery in the $ROI_b$. The imaging was acquired using the time-lapse function of the Zeiss LSM880 confocal system. The normalization of the intensity at the time point was done by the equation $(ROI_{nb}-ROI_b)/(ROI_{nb0}-ROI_{b0})$, where $ROI_{nb0}$ and $ROI_{b0}$ indicated the intensity of $ROI_{nb}$ and $ROI_b$ before bleaching. The average at each timepoint among different condensates was plotted using Prism software.

## Transmission electron microscopy (TEM) sample preparation
The morphology of the peptides was characterized using FEG-TEM, FEI Tecnai G2 TF20 Super TWIN instrument. JSF1 (3 mM) in 100 mM $K_2HPO_4/KH_2PO_4$ with or without photoinitiation was incubated at 20 °C for 24 h. For $FUS_{50-60}$, peptides were dissolved in 40% acetonitrile aqueous solution to make a 500 μM stock, diluted to 50 μM in 25 mM Tris/HCl pH 7.4 buffer and then incubated for 24 h. All samples (5 μL) were dropped onto 300 mesh copper grids and left for 30 seconds to allow the sample to attach to the grid. The copper grid was dried by absorbing the solvent from the edge of the grid with filter paper. Subsequently, the sample was negatively stained with 1% (w/v) uranyl acetate for 1 minute. The staining dye was removed, and the grid was dried inside a desiccator.

## Atomic force microscopy (AFM)
Two samples were used for AFM measurements: fibrils on HOPG and nanocondensate on mica. Initially, a drop of solution containing either fibrils or condensates (~50 μl) was deposited on a freshly cleaved HOPG (or mica) surface. After 10 min of deposition, the solution was gently removed from the HOPG/mica surface with a piece of Kimwipes wiper. The sample (either fibrils/HOPG or condensate/mica) was placed on the AFM sample stage quickly and sealed into a closed fluid cell, which was equipped with our AFM system. A buffer solution (100 mM $K_2HPO_4/KH_2PO_4$, pH 7.0, ~60 μl) was injected into the fluid cell for AFM measurement. AFM was performed with a Bruker AXS Multimode NanoScope V at room temperature (23–25 °C). PeakForce Quantitative Nano-Mechanics (PF-QNM) was employed for AFM

measurements. This operation mode provides simultaneous topographic imaging and mechanical properties mapping. Si cantilevers (PPP-FMAuD, NanoSensors) with a spring constant of ~ 2 N/m were employed. We followed the recommendation procedures of the PF-QNM user guide for calibration. The spring constant was determined by measurement of the thermal noise spectra of the cantilever and fitting with Bruker AFM software. A standard sample (PDMS-Soft-2, Bruker) with a known Young's modulus (E ~ 3.5 MPa) was used for indentation of ~ 5 nm in calibration of the tip radius. The sample's Possion ratio of 0.4 was used, as recommended by Bruker's user guide for a sample stiffness of 0.1–1 GPa. Based on the indenter geometry and our sample stiffness, the Derjaguin–Müller–Toporov (DMT) model was selected as our theoretical framework for approximation of the stiffness of the samples.

### Attenuated total reflectance Fourier transform infrared spectroscopy, ATR-FTIR (ATR-FTIR)

3 mM JSF1 1 mL with or without photoinitiation was incubated at 20 °C for 24 hours. The solution was centrifuged at 22637 × g for 1 hour at 4 °C. The supernatant was removed and the pellet was resuspended in 50% acetonitrile, frozen in liquid nitrogen and lyophilized. The powder was pressed onto the ZnSe crystal. Infrared spectra were collected on a Jasco-FT/IR-6700 spectrometer (JASCO Corporation, Tokyo, Japan) in ATR mode with a spectral resolution of 4 cm⁻¹ and wavenumber over 350–7000 cm⁻¹. The amide I region (1600–1700 cm⁻¹) was background subtracted, and deconvolution was performed by Origin2021 (OriginLab Corporation, Northampton, MA, USA) Peak Deconvolution app (Lorentzian). A second derivative was used to determine the hidden peaks, and iteration was applied.

### Time-lapse TIRF imaging

Time-lapse TIRF image collection was carried out using a Nikon TiE microscope, where samples were illuminated with a 405 nm laser light source for ThT excitation. For ThT staining, JSF1 (200 μM) was prepared in 200 mM $K_2HPO_4/KH_2PO_4$ followed by photoinitiation. The sample was mixed with the same amount of ThT (200 μM in ddH₂O) to reach the working concentration (100 μM JSF1, 100 mM $K_2HPO_4/KH_2PO_4$, 100 μM ThT). An aliquot of 200 μL was mounted on a 35 mm glass-bottom dish (Ibidi, Germany). Images were acquired for 16 h at 30 min intervals by time-lapse TIRF microscopy with a 405 nm laser. The ThT signals were filtered with an ECFP cube (Chroma) and collected by an Andor iXon3 888 back-illuminated high-sensitivity EMCCD camera.

### Protein expression, purification, and labeling

Plasmid, MBP-FUS_FL_WT was as described (Addgene plasmid # 98651; http://n2t.net/addgene:98651; RRID: Addgene_98651)[30]. The plasmid was transformed into *E. coli* BL21 (JUMBO-40 Value 107 HIT-21, RBCBioscience). His₆-MBP-TEV site-FUS was induced at 37 °C for 6 hours using 1 mM isopropyl 1-thio-β-D-galactopyranoside (IPTG). The induced cells were pelleted, resuspended in binding buffer (25 mM Tris/HCl, pH 7.4, 150 mM KCl, 4 mM β-mercaptoethanol) and broken using the cell disruption system with settings of 3 passages at 20,000 psi at 4 °C. Crude lysate was collected and centrifuged at 13,751 × g for 60 min at 4 °C. The supernatant was mixed with Ni Sephorose™ 6 Fast Flow resin overnight at 4 °C with rotation. Bound proteins were washed with binding buffer to remove the unspecific bound proteins and then eluted with elution buffer (25 mM Tris/HCl, pH 7.4, 150 mM KCl, 4 mM β-mercaptoethanol) containing 50, 100, 200, and 500 mM imidazole. The buffer of eluents containing His₆-MBP-TEV site-FUS (determined by 12% Tris–glycine SDS-PAGE) was exchanged into storage buffer (25 mM Tris/HCl, pH 7.4, 150 mM KCl, 4 mM β-mercaptoethanol, 10% v/v glycerol) by using Amicon® Ultra-15 centrifugal filter units 10,000 NMWL. To label the protein with Alexa Fluor™ 488, purified protein was incubated with Alexa Fluor™ 488 C5 Maleimide

(ThermoFisher, #A10254) at a 1:1 molar ratio in PBS (pH 8.0) and incubated at 37 °C for 1 hour with 1000 rpm shaking. The buffer was changed to storage buffer by using Amicon® Ultra-15 centrifugal filter units at 10,000 NMWL. Samples were frozen with liquid nitrogen and stored at −80 °C. Sample concentrations were estimated using the extinction coefficients calculated by ProtParam[54].

### Protein/JSF1 colocalization, LLPS promotion test and seeding assay

1 nmol of His₆-MBP-TEV site-FUS-488 was incubated with 10 units of TEV protease (New England BioLabs Inc., USA) in TEV protease reaction buffer at 30 °C for 1 hour to remove the His₆ and MBP tags. The cleaved protein was diluted into buffer (100 mM Tris/HCl, pH 7.0, 150 mM KCl) to make a 2.5 μM solution. Then, 20 μM JSF1 and 5 μM f-JSF1 were added to the protein solution and observed by confocal microscopy to confirm their colocalization. To confirm whether JSF1 could promote LLPS of FUS, 2.5 μM unlabeled FUS condensates were prepared through the same procedures, and JSF1 was added to the desired concentration (6.25–100 μM). The condensate density was monitored by DIC and analyzed by ImageJ. To conduct the seeding assay, FUS condensates (2.5 μM) were incubated in the absence or presence of JSF1 (25 μM) with or without photoinitiation, and their morphology was monitored by DIC microscopy. After 24 hours, the morphology of the protein was characterized by DIC and TEM. To prepare TEM grids, 5 μL of sample was dropped on copper grids for 30 seconds and washed with ddH₂O for 2 times. The copper grid was dried by absorbing the solvent from the edge of the grid with filter paper. Subsequently, the sample was negatively stained with 1% (w/v) uranyl acetate for 1 minute. The staining dye was removed, and the grid was dried inside a desiccator.

### Immunogold staining

50 μL 1% BSA in PBS was mixed with 50 μL of protein seeding sample for 1 hour at room temperature and then centrifuged at 16363 × g for 30 min. The supernatant was removed from the sample. 5 μL of 1:20 primary antibody in PBS (ab84078 anti-TLS/FUS antibody, Abcam, USA) was added, followed by overnight incubation at 4 °C. The sample was washed twice by adding 20 μL of PBS and centrifuged at 16363 × g for 30 min. Later, 5 μL of 1:20 secondary antibody in PBS (ab105294 Donkey Anti-Rabbit IgG H&L (6 nm Gold) preadsorbed, Abcam, USA) was added, followed by a 1 h incubation at room temperature. After incubation, the sample was washed twice by adding 20 μL of PBS and centrifuged at 16363 × g for 30 min. Then, 5 μL of the remaining liquid was dropped onto 300 mesh copper grids and left for 30 seconds to allow the sample to attach to the grid. Fixation was performed by adding 5 μL of freshly prepared 1% glutaraldehyde in PBS to the grid and left for 10 min at room temperature. Subsequently, the sample was negatively stained with 1% (w/v) uranyl acetate for 1 minute. The staining dye was removed, and the grid was dried inside a desiccator.

### Cell maintenance, transfection, peptide treatment, and photoinitiation

Mouse neuroblastoma N2A cell line was a gift from Dr. Yijuang Chern (Institute of Biomedical Sciences, Academia Sinica, Taiwan). Cells were cultured in Dulbecco's modified Eagle's medium (Invitrogen) supplemented with 2 mM glutamine, 10% heat-inactivated fetal bovine serum, and 100 U/mL penicillin–streptomycin (Invitrogen) at 37 °C in a humidified atmosphere with 5% CO₂. For FUS^R522G-EGFP expression, 2 × 10⁵ N2A cells were seeded in a 35 mm glass bottom dish and transfected with 1 μg of plasmid with Lipofectamine™ 3000 Transfection Reagent (Invitrogen # 11668019) according to the manufacturer's protocol for 2 h. After transfection, the culture medium was refreshed. JSF1 peptide was prepared by dissolving the lyophilized powder at the desired concentration in the cell culture medium. Cells were incubated with JSF1-containing medium for 6 h, photoinitiated (mercury lamp

with 345–385 nm bandpass filter, average power: 8.24 mW/cm$^2$), refreshed into new cultured medium and incubated for another 16 h.

## FRAP assay in living N2a cells

A representative FUS$^{R522G}$-EGFP condensate was selected with a-plan-Apochromat 63×/1.46 oil (Carl Zeiss), a PMT detector and ZEN 2011 software (black edition, Carl Zeiss). Cells were kept in a Zeiss Temp Module system at 37 °C and 5% CO$_2$ of the working system during the experiment. Within the investigated region (in the nucleus or cytoplasm), a circular region of interest (ROI) was taken, and 5 control images were taken before bleaching. Then, the ROI was bleached in 100 cycles in a 100% power 480 nm laser, and a series of images was captured immediately after bleaching. All quantitative analyzes were performed using ZEN 2011 software (black edition, Carl Zeiss).

## Fractionation into soluble/insoluble fractions

N2a cells were seeded in a 6-well plate at a concentration of $2 \times 10^5$ cells/well and incubated overnight. FUS$^{R522G}$-EGFP was overexpressed using Lipofectamine™ 2000 Transfection Reagent (Invitrogen, # 11668019) for 2 hours, and the culture medium was replaced with fresh medium containing 10 μM JSF1 for another 6 hours, followed by photoinitiation for 3 min and incubation for 18 hours. The cells were harvested by RIPA buffer (PBS containing 0.1% Triton-X and protease inhibitor cocktail (Roche)), sonicated on ice for 5 s, and kept on ice for 30 min with periodic vortexing. The protein concentrations of the cell lysates were determined by a Bio-Rad DC protein assay (#5000111). Extracts containing 100 μg protein were centrifuged at $20,000 \times g$ for 20 min at 4 °C. The supernatants were collected as RIPA-soluble fractions. The pellets were washed with RIPA buffer twice by centrifugation at $20,000 \times g$ for 10 min at 4 °C. The washed pellets were dissolved in PBS containing 1% sarkosyl and collected as RIPA-insoluble fractions. Proteins were separated using 12% Tris–glycine SDS-PAGE. Proteins were transferred onto PVDF membranes (Millipore). Blots were blocked with 5% bovine serum albumin (BSA, Sigma) in 0.1% PBST for at least 1 h. After blocking, blots were subjected to incubation with the primary antibodies GFP (1:1000, Abcam, ab183734) or GAPDH (1:10,000, GeneTex, GTX627408) in 2–5% BSA and incubated overnight at 4 °C on a shaker. After washing with 0.1% PBST, the blots were further incubated with HRP-labeled secondary antibodies [1:15,000, anti-Rabbit (GeneTex, GTX213110-01), anti-Mouse (Jackson ImmunoResearch Laboratories, Inc., 115-035-003)] at room temperature for another 2 h. The blots were washed and developed with electrochemiluminescence (ECL, Millipore). The signals were visualized with luminescence (iBright™ FL1000 instrument, Invitrogen). The results were analyzed by ImageJ, and the EGFP (insoluble)/GAPDH (soluble) ratio was calculated.

## Cell toxicity

A ReadyProbes® Cell Viability Imaging Kit (Blue/Red, #R37610) was added to the cells, and the sample was incubated at 37 °C for 30 min. The imaging was carried out using an automated Nikon TiE microscope. Images were collected using a 60X oil immersion objective and recorded by an Andor iXon3 888 EMCCD camera. The results were analyzed with ImageJ by calculating the red/blue signal ratio.

## Sequence analysis

FUS$_{1-165}$ was analyzed by PONDR[55], PLAAC[52], PScore[53], and PASTA 2.0[23]. For PONDR, the VLXT predictor was chosen. For PLAAC, core length 60 and Relative weighting of background probabilities (α) 100 were selected. For PASTA 2.0, top pairing was set at 20 and the energy threshold was set at −5.

## Reporting summary

Further information on research design is available in the Nature Portfolio Reporting Summary linked to this article.

## Data availability

Source data are provided with this paper.

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

## Acknowledgements

This research was supported by Academia Sinica [AS-CDA-109-M09, AS-TP-106-L13 (Joseph Jen-Tse Huang), and AS-CFII-111-302 (Ing-Shouh Hwang)] and National Science and Technology Council [NSTC 112-2113-M-001-024 and 111-2113-M-001-004 (Joseph Jen-Tse Huang)]. We thank Dr. Chih-Wen Yang (Academia Sinica) for his suggestion about AFM experiments. We thank Dr. Ruei-Yu He (Optical Microscopy Facility in Academia Sinica), Dr. Mei-Chun Tseng and Ping-Yu Lin (Mass Spectrometry Core in Academia Sinica), Dr. Po-Yen Lin and Dr. Yao-Kwan Huang (Biological Electron Microscopy Core in Academia Sinica), and Academia Sinica Neuroscience Core Facility [AS-CFII-110-101 (Ya-Jen Cheng)] for their technical supports. We thank the Data Science Statistical Cooperation Center of Academia Sinica (AS-CFII-111-215) for statistical support. Protein purification and cellular experiments were supported by Yin-Chih Hsu, Jhu-Ying Hong, Jia-Yu Chou, and Tz-Ting Chen from Acdamica Sinica. We thank Wallace Academic Editing for the revision. We also acknowledge for the critical reading and suggestions from Dr. Jun-An Chen from Academia Sinica and Dr. Jie-Rong Huang from NYCU.

## Author contributions

J.J.H. conceived the project. J.J.H. and H.Y.C. designed the peptide sequence and experiments. H.Y.C., W.T.S. and R.Y.H. characterized the LLPS and fibrilization properties. H.Y.C. and Y.J.C. performed the FLIP assay. J.J.S. and N.W. synthesized the fluorophore (JJS-0341). I.S.H. and Z.R.G. designed and performed AFM experiments. R.Y.H. and Y.A.H. performed cellular experiments. H.Y.C. and J.J.H. wrote the manuscript.

## Competing interests

The authors declare no competing interests.
