## [Peer Review File · Nature Communications]

Reviewers' Comments:

Reviewer #1:

Remarks to the Author:

The manuscript by Chuang et al. reports the design and synthesis of a photocleavable peptide able to undergo liquid-liquid phase separation (LLPS) in the dark and self-assemble into amyloid-like fibrils upon light-activation. The peptide consists of a fibre-forming portion of the FUS protein coupled to a solubilizing tag consisting of repeating arginine units through a methoxynitrobenzene cleavable moiety. This peptide is employed to promote FUS phase separation and initiate its fibrillization following light exposure. This method is ultimately applied to regulate FUS fibril formation within live cells.

Deciphering the impact of protein LLPS on the formation of amyloidogenic fibrils is a current area of interest but poses significant challenges. Developing new tools that enable control over and initiation of protein fibrillization with precise temporal resolution offers a promising avenue to gain insight into the connection between LLPS and protein aggregation. The reported peptide contributes to this endeavour and, in that regard, serves as a promising valuable tool for investigating FUS phase separation and fibril formation. Since light is bio-orthogonal and shows excellent spatiotemporal resolution, I believe this peptide provides a smart addition to the methods already available to trigger fibre formation.

The manuscript is clear and data well presented. However, a few experimental details are missing together with some key control experiments to fully back up the authors claim. The following points should particularly be addressed before publication.

1) In Figure 1, experiments were performed in the presence of PEG, but all other experiments were not. Macromolecular crowders are known to favour LLPS via excluded volume effects. Are the same UCST-like behaviour, salt dependence of JF1 phase separation, phase diagram, etc. observed in the absence of PEG? In addition, comparing the condensate density is only relevant if all images were acquired after a sufficient time after sample mounting in the observation chamber to allow all droplets to settle at the bottom of the chamber (this would take longer for smaller droplets). Was it the case? This is not specified in the methods.

2) The photocleavage of JSF1 using UV light for 3 minutes is shown to produce fibrils in Figure 2 after 24h. However, I couldn't find the control experiment showing what the system becomes after 24h without UV light irradiation. In that sense, Figure 2b is misleading as the associated legend is not clear enough: the image without UV seems to have been acquired at $t=0$ and not after 24h (the condensates should settle at the bottom of the cuvette after 24h, i.e. the two pictures would look very similar after 24h). This should be clarified and images shown at the same time without or with light exposure. Similarly: Were FLIP experiments in Figure 2c performed after 24h for both samples (without and with UV light exposure)? This again is not specified.

3) Related to this droplets-aggregate transition, were DIC images acquired with light applied in situ (i.e. on samples placed in the observation chamber) or were samples kept in tubes/cuvettes, then pipetted before imaging? Please specify the method here. I think it would be very informative to image the JSF1 droplets (both those kept in the dark and those exposed to UV) overtime. This would allow to monitor the transition from droplets to aggregates in situ. In particular, it would be interesting to know what the kinetics of the process is. Why do fibrils only emerge after 24h? What is the limiting step if not the photocleavage step (UV light is only applied for 3 minutes)?

4) Regarding the impact of JSF1 on FUS phase separation and fibrillization: on Figure 3a, it looks like FUS condensates become less spherical and tend to cluster when increasing concentrations of JSF1 are added. Could the authors comment on that? Also, why did the authors explore such high concentrations of JSF1 since in the rest of the study they only use 2.5 μM of the peptide? Other point, on page 8, line 155: from the fluorescence microscopy images, one can only conclude that f-

JSF1 is partitioned in the FUS condensate. However, unlabelled JSF1 could behave differently. The authors should be more cautious in their conclusions unless they can determine the amount of unlabelled JSF1 sequestered in the FUS condensates.

I also have a few minor comments:

5) Some other details should be added:

- page 4, line 68: what is the ionic strength of the phosphate buffer used (add in the main text)?
- page 4, line 68: please specify here the amount of PEG used.
- Fig. S2: What do the two and single star mean?
- Fig. S12: there is no (a) and (b)
- Fig. S15: What are those images? Epifluorescence? Confocal microscopy?

6) Fig. S6: please provide a full-width UV/vis spectrum of JJSF1 (irradiation is performed at 365 nm, so please show the spectra down to 200 nm). How come the baseline at wavelengths > 500 nm does not reach zero? Are samples turbid?

7) Some terminology is not clear and should be revised or specified:

- page 3, line 33: what does "more viscoelastic" mean? More viscous? More elastic?
- page 5, line 85: "the shielding effect" is an inaccurate concept to describe charge screening by salt.
- The term "photoinitiation" could gain in clarity in some places if used to designate what light actually does to the system, such as "peptide photocleavage"

8) I spotted a few mistakes:

- page 2, line 18: "facilitated" should be "facilitate"
- page 4, line 53: "...conjugating an FUS..." should be "...conjugating a FUS..."
- Fig. S15: "colocalize" should be "colocalization"

Reviewer #2:

Remarks to the Author:

Summary: In this manuscript by Chuang et al, the authors describe a photo-cleavable peptide containing a FUS LCD peptide (YGQSSYSSYGQ) and Arg6 motif. The peptide itself undergoes UCST-type phase separation and forms liquid-like droplets at mM concentration range. When Arg6 is cleaved, these droplets then form amyloid fibers. The peptide can interact with FUS in vitro and in cells and undergo co-condensation. Upon similar photocleavage, fibers were observed for FUS+peptide systems in vitro but in cells, the authors did not observe such liquid-to-solid transition. Instead, they observed FRAP recovery of FUS in the dense phase to go down. Cell toxicity was observed to modestly increase upon photoinitiations, but interestingly, it decreased in presence of the peptide under dark conditions. Based on these experimental data, the authors argue that they have engineered a phase modulator peptide for FUS that undergo phase separation but transform into amyloid-like fibrils upon photoinitiation.

Major points:

1. The peptide design works as the authors have shown experimentally but the design principles are lacking and questionable. There has not been any discussion how the authors came about designing this particular peptide. Why Arg6 motif was added to a FUS LCD peptide? Why is this specific FUS LCD peptide uses? It is very clear what is happening in this system: the YGQSSYSSYGQ motif is a steric zipper (one can check this here: <https://zipperdb.mbi.ucla.edu/>) and templates amyloid formation and Arg6 likely interacts with the Tyr residues and results in phase separation; removal of Arg6 frees the zipper motifs to form amyloid fibers. The Arg6 motif maybe required for cell penetration.

2. What is the evidence that upon photoinitiation, the FUS+peptide system fibers contain FUS? They could just be homotypic peptide fibers.

3. The mechanism of FUS phase separation and "proposed" amyloid formation needs further quantification. Does the Arg6 lowers the C_{sat} of FUS or is it the zipper motif alone? After photoinitiation and Arg6 cleavage, what happens to the Arg6 motif? Does it non-covalently interact with FUS and the FUC LCD peptide?
4. The description of the mechanical properties of peptide condensates and their fibers are too vague. What does DMT modulus even mean? Are there reference samples that can help the readers to interpret what these numbers supposed to be for liquid and solid-like condensates? Just reporting a measured quantity without describing what it is without proper controls and meaning can be problematic.
5. All experiments should be repeated with Arg6 and the FUS LCD peptide as a control, including cellular experiments.
6. What is the advantage of the photo-activatable cleavage of Arg6 while a similar outcome can be obtained using a protease? Here is a work where protease cleavable phase separating constructs were used.
7. Why does cellular FUS condensate not form amyloid fibers?
8. The term phase modulator is misleading, it's a peptide ligand that nucleates FUS phase separation and it's the zipper motif that acts a steric zipper for FUS aggregation, although it is not clear why the FUS condensates do not form fibers in the cell or even the in vitro sample showing fibers contain FUS. This is where further studies as indicated in point # 5 will be highly useful.
9. Why FUS condensates are less toxic in presence of the peptide?
10. The readability of the paper can be improved in several instances, the results were discussed in one place and the readers are asked to check discussion sections for the interpretation of the data.
11. How does the current study improve our "understanding the molecular mechanism underlying the condensation and maturation of biomolecular condensates is crucial for delineating the physiology and pathology of various biological processes"
12. Can the authors characterize the time evolution of samples by imaging after photoinitiation to map the pathway for droplet to fiber transition? Currently, movie 2 shows fiber formation only after 12-16 hours, but that does not provide any picture of how photoinitiation resulted in droplet to fiber formation.

Minor point:

1. Why is the peptide called JSF1?
2. Why does the condensates turn yellow upon photoinitiation?

Reviewer #3:

Remarks to the Author:

In this article, the authors successfully engineered photo-controllable peptides to modulate the liquid-liquid phase separation of FUS. Using a combination of biophysical methods, this engineered peptide, JSF1, was found not only to phase separate by itself but also capable of regulating the fluidity of FUS droplets in vitro and in vivo. This article's findings help pinpoint a vital region that drives the maturation of FUS LLPS and have provided a powerful tool that may benefit future studies in this field.

To further improve this article, here are my comments:

1. During the turbidity assay, the temperature was ramped up at the rate of 5°C/min, and the critical temperature of phase separation was quantified based on this measurement. Thus, whether the system has reached its steady state must be confirmed. Otherwise, if the system cannot stabilize itself as quickly as the temperature ramps up, the measurements cannot reflect the actual temperature threshold of phase separation.
2. A 405nm laser, close to the photoinitiation wavelength of 365nm, was used in the FLIP experiments. Meanwhile, according to the absorption data in Figure S6, JSF1 has a non-zero absorbance at around 400nm. In this case, it would be better to provide some evidence showing the impact of the bleaching lasers on the cleavage of JSF1.

3. In the discussion part, it was mentioned that the LCD fragments and poly-R contributed to the interactions that "promote LLPS." However, the results showed increased fluidity and a change in number density. These influences are not precisely equal to the promotion of LLPS. Thus, the expression here might need some adjustment, or more evidence, such as the change of the critical temperature, salt concentration, or FUS concentration under the impact of JSF1, needs to be provided.

4. The cell-based assays were performed with FUS-transfected cells only. Without the controls with the non-transfected conditions, it would be hard to determine the effect of the JSF1 treatment alone and the effect of light cleavage alone to the cell toxicity. Thus, it's essential to include the controls with -FUS-eGFP, + JSF1 and +/- light cleavage.

We appreciate for all the comments and suggestions from the reviewers. The detailed responses to each reviewer are provided below.

Response to Reviewer 1:

The manuscript by Chuang et al. reports the design and synthesis of a photocleavable peptide able to undergo liquid-liquid phase separation (LLPS) in the dark and self-assemble into amyloid-like fibrils upon light-activation. The peptide consists of a fibre-forming portion of the FUS protein coupled to a solubilizing tag consisting of repeating arginine units through a methoxynitrobenzene cleavable moiety. This peptide is employed to promote FUS phase separation and initiate its fibrillization following light exposure. This method is ultimately applied to regulate FUS fibril formation within live cells.

Deciphering the impact of protein LLPS on the formation of amyloidogenic fibrils is a current area of interest but poses significant challenges. Developing new tools that enable control over and initiation of protein fibrillization with precise temporal resolution offers a promising avenue to gain insight into the connection between LLPS and protein aggregation. The reported peptide contributes to this endeavour and, in that regard, serves as a promising valuable tool for investigating FUS phase separation and fibril formation. Since light is bio-orthogonal and shows excellent spatiotemporal resolution, I believe this peptide provides a smart addition to the methods already available to trigger fibre formation.

The manuscript is clear and data well presented. However, a few experimental details are missing together with some key control experiments to fully back up the authors claim. The following points should particularly be addressed before publication.

Comment 1: In Figure 1, experiments were performed in the presence of PEG, but all other experiments were not. Macromolecular crowders are known to favour LLPS via excluded volume effects. Are the same UCST-like behaviour, salt dependence of JPFSI phase separation, phase diagram, etc. observed in the absence of PEG? In addition, comparing the condensate density is only relevant if all images were acquired after a sufficient time after sample mounting in the observation chamber to allow all droplets to settle at the bottom of the chamber (this would take longer for smaller droplets). Was it the case? This is not specified in the methods.

Response: We appreciated for this comment and have now added new experiments to identify the LLPS property of JSF1 without PEG. Our results showed that the biophysical property of JSF1 remained similar with or without PEG. The new data without PEG could be found in the revised manuscript (**Fig. 1c-1e**) and in the revised supplementary information (**Supplementary Fig. 3, 5, and 6**).

In the revised manuscript (**page 5, line 75-95**), we have now added: “Differential interference contrast (DIC) microscopy was employed to monitor the room-temperature LLPS of JSF1 at various concentrations in phosphate buffer (100 mM K₂HPO₄/KH₂PO₄, pH 7.0; details in the Methods section). Our results showed that spherical JSF1 droplets formed when the JSF1 concentration was 1 mM or higher (Fig. 1c). The density and diameter of these droplets were positively correlated with the JSF1 concentration (Fig. 1d and Supplementary Fig. 3) ... To map the phase diagram of JSF1 in terms of its concentration and the temperature, we measured the turbidity of JSF1 at various temperatures. The results revealed that decreasing the temperature from 40 to 5 °C considerably increased the turbidity of JSF1 solutions of all JSF1 concentrations (0.5–5 mM; Supplementary Fig. 5a), ... It is worth to note that JSF1 condensation could be enhanced by the crowding agent PEG₈₀₀₀ (0–30%, Supplementary Fig. 7) and the fusion events could be clearly observed (Fig. 1f and Supplementary Video 1).”

We also appreciated for the suggestion of controlling the acquisition time when comparing the condensate density. More details are now added in the revised manuscript as shown below. In addition, we have added new experiment to confirm the concentration dependency by measuring the turbidity of JSF1 at different conditions (**Supplementary Fig. 5a**).

In the revised manuscript (**page 15, line 323-325**), we have now added: “DIC images of condensates were taken after 10 min of incubation after samples mounted on the glass-bottom dish and further analyzed by ImageJ. The condensates were manually counted, and the diameters were measured.”

Comment 2: The photocleavage of JSF1 using UV light for 3 minutes is shown to produce fibrils in Figure 2 after 24h. However, I couldn't find the control experiment showing what the system becomes after 24h without UV light irradiation. In that sense, Figure 2b is misleading as the associated legend is not clear enough: the image without UV seems to have been acquired at t=0 and not after 24h (the condensates should settle at the bottom of the cuvette after 24h, i.e. the two pictures would look very similar after 24h). This should be clarified and images shown at the same time without or with light

exposure. Similarly: Were FLIP experiments in Figure 2c performed after 24h for both samples (without and with UV light exposure)? This again is not specified.

Response: We appreciated for this question and have now added new control experiments to address the aforementioned issue accordingly (**Fig. 2b and 2c**). We have also added the details for FLIP experiments in Figure 2d (**page 7, line 123-124**). The incubation time for both samples (without and with UV light exposure) were 24 h.

In the revised manuscript (**page 6, line 106-111**), we have now added: “The JSF1 solution (3 mM) was cloudy and contained spherical condensates under differential interference contrast microscopy (Fig. 2b). After 24 h of incubation, these condensates settled down to the bottom of the tube and make the solution transparent (Fig. 2b). Upon photocleavage, the solution turned yellow at 0.5 h and the spherical morphology of these condensates persisted (Fig. 2c). After further incubation, brown depositions were found at the bottom of the tube and irregular aggregates were revealed (Fig. 2c) at 24 h.”

In the revised manuscript (**page 7, line 123-124**), we have added: “After 24 h of incubation, the samples with or without photoinitiation were applied to FLIP assay.”

Comment 3: Related to this droplets-aggregate transition, were DIC images acquired with light applied in situ (i.e. on samples placed in the observation chamber) or were samples kept in tubes/cuvettes, then pipetted before imaging? Please specify the method here. I think it would be very informative to image the JSF1 droplets (both those kept in the dark and those exposed to UV) overtime. This would allow to monitor the transition from droplets to aggregates in situ. In particular, it would be interesting to know what the kinetics of the processis. Why do fibrils only emerge after 24h? What is the limiting step if not the photocleavage step (UV light is only applied for 3 minutes)?

Response: Due to the high affinity between JSF1 and glass, the longer incubation perturbed the morphological observation during our experiments (data not shown). Therefore, samples were kept in tubes then pipetted before imaging. The related details are now added in the Methods section of the revised manuscript (**page 15, line 319-320**). As suggested by Reviewer #1, we have now added new experiment to image the JSF1 droplets (both those kept in the dark and those exposed to UV) overtime (**Supplementary Fig. 9**).

In the revised manuscript (**page 6, line 111-113**), we have now added: “Additionally, time-course differential interference contrast microscopy (Supplementary Fig. 9) demonstrated that the photoinitiated condensates became non-spherical after 14 h of incubation and gradually transformed into aggregates.”

In the revised manuscript (**page 15, line 319-320**), we have added: “An aliquot of 200 μ L was transferred from Eppendorf to a 35 mm glass-bottom dish (Ibidi, Germany) before imaging.”

As for the question related to the fibrilization kinetics, due to the resolution limitation of DIC and instrument-response time, it is hard for us to answer now. In fact, fibers were generated even during the irradiation and the photocleaved JSF1 aggregates in Figure 2 consisted of huge amounts of fibers. Therefore, to address this issue, time-course experiment based on molecular resolution with ultra-fast instrument-response time (e.g., light sheet fluorescence microscopy) are needed. Currently, we do not have the related resources.

Comment 4: Regarding the impact of JSF1 on FUS phase separation and fibrillization: on Figure 3a, it looks like FUS condensates become less spherical and tend to cluster when increasing concentrations of JSF1 are added. Could the authors comment on that? Also, why did the authors explore such high concentrations of JSF1 since in the rest of the study they only use 2.5 μ M of the peptide? Other point, on page 8, line 155: from the fluorescence microscopy images, The authors sone can only conclude that f-JSF1 is partitioned in the FUS condensate. However, unlabelled JSF1 could behave differently. should be more cautious in their conclusions unless they can determine the amount of unlabelled JSF1 sequestered in the FUS condensates.

Response: Currently, we surmise the morphological change together with the clustering of the FUS condensates are due to the partial fusing events of FUS condensates by increasing JSF1 molecules. In fact, similar phenomenon has also been reported in other published work either with or without additional modulators (Babinchak et al., 2020; Babinchak et al., 2019).

We also appreciated for the question of the various concentration of JSF1. This is mainly due to the interesting biophysical properties of this molecule. At low concentration (25 μ M), JSF1 could not form condensates but was able to promote the LLPS of purified FUS protein. Meanwhile, JSF1 itself could also form condensates at

higher concentration (3 mM). To make this point clear, we have now added some new control experiments and revised **Fig. 3a-c** accordingly.

In the revised manuscript (**page 9, line 164-171**), we have now added: “After the His₆-MBP tag was removed from the recombinant protein, FUS formed spherical condensates at the concentration equal or higher than 1.25 μM (Fig. 3a and Supplementary Fig. 17c). To determine whether JSF1 could modulate the LLPS of FUS, we incubated FUS condensates (2.5 μM) with various concentrations of JSF1 (6.25–100 μM). Note that JSF1 alone did not form condensates at these concentrations (Supplementary Fig. 18). Our results showed that the density of FUS condensates was increased by JSF1 (Fig. 3b and 3c) and the critical concentration of FUS LLPS could be reduced by JSF1 (Supplementary Fig. 19), suggesting JSF1 could facilitate FUS LLPS.”

As for the partition of JSF1, we believe both the unlabeled JSF1 and *f*-JSF1 partitioned in the FUS condensates. As shown in the revised **Supplementary Fig. 20**, we had spiked trace *f*-JSF1 into large amount of unlabeled JSF1 (ratio 1:2500) to minimize the impacts of fluorophore and still found the colocalization between JSF1 and FUS. In addition, we had also added a new centrifugation experiment to show that around 67% of unlabeled JSF1 could be sequestered into purified FUS condensates (details in **Fig. M1 of Materials for reviewer only**).

Comments 5: Some other details should be added:

- *page 4, line 68: what is the ionic strength of the phosphate buffer used (add in the main text)?*

- *page 4, line 68: please specify here the amount of PEG used.*

- *Fig. S2: What do the two and single star mean?*

- *Fig. S12: there is no (a) and (b)*

Fig. S15: What are those images? Epifluorescence? Confocal microscopy?

Response: We appreciated for these comments. In the revised manuscript (**page 5, line 76-77**), we have now added: “Differential interference contrast (DIC) microscopy was employed to monitor the room-temperature LLPS of JSF1 at various concentrations in phosphate buffer (100 mM K₂HPO₄/KH₂PO₄, pH 7.0; details in the Methods section)”

In the revised manuscript (**page 6, line 94**), we have added: “...the crowding agent PEG₈₀₀₀ (0–30%, Supplementary Fig. 7) ...”

In the revised supplementary information (**page 10, Supplementary Fig. 3**), we have now added: “Data were analyzed by one-way ANOVA with Tukey post-hoc test (***) $P < 0.001$, n.s. non-significant).”

In the revised supplementary information (**page 15, Supplementary Fig. 15**), we have now removed the (a) and (b).

In the revised supplementary information (**page 16, Supplementary Fig. 20**), we have now specified the images were acquired by epifluorescence.

Comments 6: Fig. S6: please provide a full-width UV/vis spectrum of JJSF1 (irradiation is performed at 365 nm, so please show the spectra down to 200 nm). How come the baseline at wavelengths > 500 nm does not reach zero? Are samples turbid?

Response: The full-width UV/vis spectrum of JSF1 is now provided (**Revised Supplementary Fig. 8a**). As shown in the new spectrum (down to 200 nm), the baseline at wavelengths > 500 nm reaches zero.

*Comments 7: Some terminology is not clear and should be revised or specified:
- page 3, line 33: what does “more viscoelastic” mean? More viscous? More elastic? -
page 5, line 85: “the shielding effect” is an inaccurate concept to describe charge screening by salt.
- The term “photoinitiation” could gain in clarity in some places if used to designate what light actually does to the system, such as “peptide photocleavage”*

Response: We have now revised and specified the aforementioned terminologies. As reported by Michieletto et al. and Shen et al., the condensates behave more like a gel during the maturation process. At this condition, both the “viscosity” and “elasticity” of the aging condensates increased with time, demonstrating the viscoelastic property (Michieletto & Marena, 2022; Shen et al., 2023).

In the revised manuscript (**page 5, line 91-93**), we have now added: “Additionally, we confirmed that the LLPS of JSF1 was suppressed (Supplementary Fig. 6) at higher KCl concentration, suggesting electrostatic interactions such as cation- π and dipole-dipole interactions were reduced at stronger ionic strength” The related description of shielding effect is now removed.

We agree with the reviewer and have now replaced “photoinitiation” by “photocleavage” accordingly (**page 6, line 109**).

Comments 7: I spotted a few mistakes:

- page 2, line 18: “facilitated” should be “facilitate”

- page 4, line 53: “...conjugating an FUS...” should be “...conjugating a FUS...”

- Fig. S15: “colocalize” should be “colocalization”

Response: We appreciated for the suggestion. The related corrections could now be found in the revised manuscript highlighted in yellow.

Response to Reviewer 2:

In this manuscript by Chuang et al, the authors describe a photo-cleavable peptide containing a FUS LCD peptide (YGQSSYSSYGQ) and Arg6 motif. The peptide itself undergoes UCST-type phase separation and forms liquid-like droplets at mM concentration range. When Arg6 is cleaved, these droplets then form amyloid fibers. The peptide can interact with FUS in vitro and in cells and undergo co-condensation. Upon similar photocleavage, fibers were observed for FUS+peptide systems in vitro but in cells, the authors did not observe such liquid-to-solid transition. Instead, they observed FRAP recovery of FUS in the dense phase to go down. Cell toxicity was observed to modestly increase upon photoinitiations, but interestingly, it decreased in presence of the peptide under dark conditions. Based on these experimental data, the authors argue that they have engineered a phase modulator peptide for FUS that undergo phase separation but transform into amyloid-like fibrils upon photoinitiation.

Comment 1: The peptide design works as the authors have shown experimentally but the design principles are lacking and questionable. There has not been any discussion how the authors came about designing this particular peptide. Why Arg6 motif was added to a FUS LCD peptide? Why is this specific FUS LCD peptide uses? It is very clear what is happening in this system: the YGQSSYSSYGQ motif is a steric zipper (one can check this here: <https://zipperdb.mbi.ucla.edu/>) and templates amyloid formation and Arg6 likely interacts with the Tyr residues and results in phase separation; removal of Arg6 frees the zipper motifs to form amyloid fibers. The Arg6 motif maybe required for cell penetration.

Response: We appreciated for these comments. Base on the suggestion, we have now added our design in the results section of our revised manuscript and rewrite our discussion section accordingly.

In the revised manuscript (**page 4, line 64-71**), we have added: “The phase modulator, JSF1, was mainly composed of FUS LCD fragment and a polyarginine tract (RRRRRR). By applying different protein aggregation predictors, we identified the protein segments (FUS₅₀₋₆₀: YGQSSYSSYGQ) in FUS LCD with high propensity for fibrilization (Supplementary Fig. 1). The polyarginine tract was select here for its cell penetrating ability. Meanwhile, we also surmise the multivalent interactions (e.g., cation- π and π - π interactions) between positively charged arginines and the three tyrosines in FUS₅₀₋₆₀

could benefit greatly on the droplet formation. To further enable the photocontrollable ability of this phase modulator, the photocleavable linker was applied to conjugate FUS LCD fragment with the polyarginine tract.”

In the revised manuscript (**page 14, line 275-283**), we have also added: “We surmised that the FUS LCD fragment (FUS₅₀₋₆₀) and the cationic polyarginine tract significantly contribute to these multivalent interactions which are essential for LLPS. In addition, through the use of various bioinformatics tools (e.g., PONDR, PLAAC, PScore, PASTA 2.0, and ZipperDB), FUS₅₀₋₆₀ was a highly disordered sequence (Supplementary Fig. 27a) that has prion-like properties (Supplementary Fig. 27b), strong π - π interactions (Supplementary Fig. 27c), and a high tendency to form both steric zipper and aggregate (Supplementary Fig. 1a and 1b). These features likely account for the amyloid-like properties (Fig. 2) and seeding capacity (Fig. 3f-3h) of the photocleaved FUS₅₀₋₆₀ peptide from JSF1.”

Comment 2: What is the evidence that upon photoinitiation, the FUS+peptide system fibers contain FUS? They could just be homotypic peptide fibers.

Response: To confirm FUS protein is contained in the FUS+peptide system, we had applied immunogold staining with anti-FUS antibody (details in Methods) in the presence or absence of JSF1 with or without photoinitiation (**Fig. 3h and Supplementary Fig. 22**).

In the revised manuscript (**page 10, line 189-192**), we have added: “Additionally, we performed transmission electron microscopy and immunogold labeling with FUS antibody to confirm that photoinitiated JSF1 promoted the formation of FUS-specific fibrils (Fig. 3h and Supplementary Fig. 22).”

Comment 3: The mechanism of FUS phase separation and “proposed” amyloid formation needs further quantification. Does the Arg6 lowers the C sat of FUS or is it the zipper motif alone? After photoinitiation and Arg6 cleavage, what happens to the Arg6 motif? Does it non-covalently interact with FUS and the FUC LCD peptide?

Response:

In order to know whether Arg6 motif could lower the saturation concentration of FUS, we have now monitored the condensate density of FUS in the presence or absence of Arg6. As shown in the revised manuscript, the saturation concentration of FUS was reduced from 1.25 μ M (**Fig. 3a**) to 0.625 μ M (**Supplementary Fig. 19**). From the

HPLC analysis (**Supplementary Fig. 8b**) and the new DIC image (**Supplementary Fig. 4a**), we learned Arg6 was successfully cleaved from JSF1 and Arg6 alone could not form droplet. Since the droplets from photoinitiated JSF1 could still maintain for 12 h, we surmise Arg6 could interact with FUS LCD peptide and/or FUS protein through non-covalent interactions.

In the revised manuscript (**page 9, line 166-173**), we have now added: “To determine whether JSF1 could modulate the LLPS of FUS, we incubated FUS condensates (2.5 μM) with various concentrations of JSF1 (6.25–100 μM). Note that JSF1 alone did not form condensates at these concentrations (Supplementary Fig. 18). Our results showed that the density of FUS condensates was increased by JSF1 (Fig. 3b and 3c) and the critical concentration of FUS LLPS could be reduced by JSF1 (Supplementary Fig. 19), suggesting JSF1 could facilitate FUS LLPS. Interestingly, poly arginine tract could also reduce the critical concentration of FUS (Supplementary Fig. 19), confirming the importance of arginines in providing the multivalency required for LLPS.”

Comment 4: The description of the mechanical properties of peptide condensates and their fibers are too vague. What does DMT modulus even mean? Are there reference samples that can help the readers to interpret what these numbers supposed to be for liquid and solid-like condensates? Just reporting a measured quantity without describing what it is without proper controls and meaning can be problematic.

Response: We appreciated this comment. We used the Derjaguin–Müller–Toporov (DMT) model to calculate the stiffness of the samples, as explained in the Methods section. The stiffness is also called “Young's modulus” or “DMT modulus”. In the revised manuscript (**page 8, line 139 and 141**), we have replaced “DMT modulus” with “Young's modulus” or “stiffness” for clarity.

As for the question related to the Yon's modulus, we calibrated the tip with a standard sample (PDMS-Soft-2, Bruker) with a known Young's modulus before measuring the stiffness of the samples with PF-QNM, as suggested by the manufacturer. This was mentioned in the Methods section of the previous manuscript. The results show that the fibrils are about five times stiffer than the condensates. We note that the low stiffness alone does not indicate that the condensates are liquid-like. However, the low stiffness (**Supplementary Fig. 13a and 13c**) and smooth surface morphology (**Fig. 2f**) are consistent with a liquid-like state. For comparison, high-density polyethylene (HDPE) has a Young's modulus of ~ 300 Mpa, which is similar to the stiffness of the fibrils (**Supplementary Fig. 13b and 13c**).

Comment 5: All experiments should be repeated with Arg6 and the FUS LCD peptide as a control, including cellular experiments.

Response: We have now prepared Arg6 and the FUS LCD peptide as new controls for both in vitro and cellular experiments. The characterization of these newly synthesized Arg6 and the FUS LCD peptides **are shown in Supplementary Fig. 2b and 2c**. Based on these new data, the manuscript has also been revised accordingly.

In the revised manuscript (**page 5, line 80-81**), we have now added: “By contrast, neither poly arginine tract nor FUS₅₀₋₆₀ could form droplets (Supplementary Fig. 2b–c and Fig. 4).”

In the revised manuscript (**page 6, line 113-116**), we have now added: “Note that poly arginine tract did not form aggregates after 24 h of incubation (Supplementary Fig. 10). On the contrary, FUS₅₀₋₆₀ formed huge aggregates in buffer (Supplementary Fig. 4), indicating the aggregates of photoinitiated JSF1 were formed by FUS₅₀₋₆₀.”

In the revised manuscript (**page 9, line 171-173**), we have now added: “Interestingly, poly arginine tract could also reduce the critical concentration of FUS (Supplementary Fig. 19), confirming the importance of arginines in providing the multivalency required for LLPS.”

In the revised manuscript (**page 10, line 188-189**), we have now added: “Note that poly arginine tract failed to trigger the liquid-to-solid phase transition of FUS (Supplementary Fig. 21), indirectly reflecting the importance of FUS₅₀₋₆₀.”

In the revised manuscript (**page 11, line 209-211**), we have now added: “FUS^{R522G}–EGFP readily formed cytosolic condensates that colocalized with *f*-JSF1 (Fig. 4b and 4c) but not fluorophore-attached poly arginine tract (Supplementary Fig. 24).”

Comment 6: What is the advantage of the photo-activatable cleavage of Arg6 while a similar outcome can be obtained using a protease? Here is a work where protease cleavable phase separating constructs were used.

Response:

Based on the information from Reviewer #2, we found the reference from Schuster et al. (Schuster et al., 2018). In this work, they demonstrated enzymatically-triggered

droplet assembly and disassembly as well as controlling of droplet composition. However, in order to create enzymatic trigger in cells, co-transfection of TEV protease along with TEV-cleavable RGG-cargo-RGG are needed. While the protease trigger provides an excellent tool for the synthetic organelle bioengineering, the aforementioned strategy is not suitable for the disease-related studies due to the potential impact and/or system disturbance from the exogenous protease. By contrast, the photo-activatable cleavage can benefit on pathological studies as photoactivation allows bio-orthogonal and spatiotemporal control of the phase transition of condensates.

In the revised manuscript (**page 12, line 244-250**), we have now added: “As recent literature has disclosed a potential correlation between the dysregulated LLPS and various diseases, extensive efforts have been made in developing different methods for regulating protein phase transitions within cells. Some approaches have focused on modulating the phase separation of RNA-binding proteins using the Cry2-based optogenetic platform, enzymatically-triggered system...”

Comment 7: Why does cellular FUS condensate not form amyloid fibers? The term phase modulator is misleading, it's a peptide ligand that nucleates FUS phase separation and it's the zipper motif that acts a steric zipper for FUS aggregation, although it is not clear why the FUS condensates do not form fibers in the cell or even the in vitro sample showing fibers contain FUS. This is where further studies as indicated in point # 5 will be highly useful.

Response: We need to first clarify that photoinitiated JSF1 could induce the amyloid fiber formation of FUS protein in vitro. As mentioned in our response for Comment #2, anti-FUS antibody was applied in our immunogold staining experiments. We demonstrated these induced fibers were composed of FUS protein by TEM images (Fig. 3h). We also need to clarify that it is difficult to observe amyloid fiber in cell through confocal imaging based on the limitation on optical resolution (the average width for FUS fiber is around 10 nm). Therefore, the cellular FUS aggregates transformed from condensates was analyzed by western blotting (details in **page 11, line 220-223 of Revised Manuscript**). The results showed that photoinitiated JSF1 led to increasing FUS amyloid aggregates in cells (**Supplementary Fig. 26**). In addition, the biophysical properties of these FUS aggregates were characterized by FRAP, which exhibited a dramatic decrease in fluidity. In fact, the non-zero FRAP of FUS aggregates in cell has also been reported in many important works (Gu et al., 2020; Li et al., 2022; Liu et al., 2020). Since JSF1 could nucleate FUS phase separation and its zipper motif could cause FUS aggregation, we surmise the “phases” of FUS protein have been modulated. The

similar concept has also been proposed by Babinchak et al. and Mitrea et al in different studies.(Babinchak et al., 2020; Mitrea et al., 2022).

Comment 8: Why FUS condensates are less toxic in presence of the peptide?

Response: Currently, the detailed correlation between FUS condensation, aggregation, and pathogenesis are still not totally understood. However, based on the reported literature, the aggregation of FUS in stress granules is strongly correlated with FUS proteinopathy (Murakami et al., 2015). We thus surmise FUS condensates are less toxic in presence of the peptide since JSF1 could increase the mobility of FUS and thus prevents its aggregation.

Comment 9: The readability of the paper can be improved in several instances, the results were discussed in one place and the readers are asked to check discussion sections for the interpretation of the data.

Response:

We appreciated for this comment and have now revised the manuscript accordingly to increase the readability.

In the revised manuscript (**page 5, line 91-95**), we have now added “Additionally, ...suggesting electrostatic interactions such as cation- π and dipole-dipole interactions were reduced at stronger ionic strength. It is worth to note that JSF1 condensation could be enhanced by the crowding agent PEG₈₀₀₀ (0–30%, Supplementary Fig. 7) and the fusion events could be clearly observed (Fig. 1f and Supplementary Video 1).”

In the revised manuscript (**page 6, line 111-116**), we have added “...Note that poly arginine tract did not form aggregates after 24 h of incubation (Supplementary Fig.10). On the contrary, FUS₅₀₋₆₀ formed huge aggregates in buffer (Supplementary Fig. 4), indicating the aggregates of photoinitiated JSF1 were formed by FUS₅₀₋₆₀.”

In the revised manuscript (**page 9, line 164-173**), we have added “After the His₆-MBP tag was removed from the recombinant protein, ...Our results showed that the density of FUS condensates was increased by JSF1 (Fig. 3b and 3c) and the critical concentration of FUS LLPS could be reduced by JSF1 (Supplementary Fig. 19), suggesting JSF1 could facilitate FUS LLPS. Interestingly, poly arginine tract could also

reduce the critical concentration of FUS (Supplementary Fig. 19), confirming the importance of arginines in providing the multivalency required for LLPS.”

In the revised manuscript (**page 10, line 188-192**), we have added “Note that poly arginine tract failed to trigger the liquid-to-solid phase transition of FUS (Supplementary Fig. 21), indirectly reflecting the importance of FUS₅₀₋₆₀...”

In the revised manuscript (**page 11, line 226-228**), we have added “Based on the FRAP analysis and western blotting, we showed that only the photoinitiated JSF1 could trigger the reduction of FUS fluidity and increase the aggregation of FUS.”

Comment 10: How does the current study improve our “understanding the molecular mechanism underlying the condensation and maturation of biomolecular condensates is crucial for delineating the physiology and pathology of various biological processes”

Response:

As growing evidence have proposed the maturation of intrinsically disorder proteins (e.g., FUS and TDP-43) condensates were closely correlated with neurodegenerative diseases, effective tools for modulating protein phases in live cells were urgently needed. With the photo-initiated toolbox in hand, we could compare the roles of “FUS condensation” in its physiological condition with that of under pathological condition. The possible impact of “FUS gelation” on other stress granules-related proteins could also be explored in the future. Moreover, the concept of “condensate-modifying therapeutics” has just been proposed by Mitrea et al (Mitrea et al., 2022). We believe our study could also benefit on therapeutics innovation by screening potential molecules that could modulate condensate-formation in diseases.

Comment 11: Can the authors characterize the time evolution of samples by imaging after photoinitiation to map the pathway for droplet to fiber transition? Currently, movie 2 shows fiber formation only after 12-16 hours, but that does not provide any picture of how photoinitiation resulted in droplet to fiber formation.

Response:

We appreciated for this comment and have now added a time-course DIC imaging demonstrating the evolution of condensates. Based on the new results, the condensates persisted until 12 h and gradually matured into aggregates after 14 h of incubation (**Supplementary Fig. 9**).

In the revised manuscript (**page 6, line 111-113**), we have now added: “Additionally, time-course differential interference contrast microscopy (Supplementary Fig. 9) demonstrated that the photoinitiated condensates became non-spherical after 14 h of incubation and gradually transformed into aggregates.”

Minor point:

- 1. Why is the peptide called JSF1?*
- 2. Why does the condensates turn yellow upon photoinitiation?*

Response:

- JSF1 is named by the first name initial of the three authors including **J**oseph Jen-Tse Huang, **S**teven Hao-Yu Chuang, and **F**inn Wan-Ting Hsu.
- The photolysis of nitrobenzene-based molecules will result in the formation of a nitrosobenzene moiety with their absorbance properties been reported before (Mary Wilcox, 1990).

Response to Reviewer 3:

In this article, the authors successfully engineered photo-controllable peptides to modulate the liquid-liquid phase separation of FUS. Using a combination of biophysical methods, this engineered peptide, JSF1, was found not only to phase separate by itself but also capable of regulating the fluidity of FUS droplets in vitro and in vivo. This article's findings help pinpoint a vital region that drives the maturation of FUS LLPS and have provided a powerful tool that may benefit future studies in this field.

Comment 1: During the turbidity assay, the temperature was ramped up at the rate of 5 °C/min, and the critical temperature of phase separation was quantified based on this measurement. Thus, whether the system has reached its steady state must be confirmed. Otherwise, if the system cannot stabilize itself as quickly as the temperature ramps up, the measurements cannot reflect the actual temperature threshold of phase separation.

Response: We appreciated for this comment and have now added new experiments to measure the turbidity at slower rate (1 °C/min). Under the new condition, the system reached its steady state and stabilized during the measurement (**Fig. 1e** and **Supplementary Fig. 5**).

In the revised manuscript (**page 15 line 332-333**), we have now added: “...For temperature-dependent measurements, the starting temperature was 5 °C and raised with a gradient of 1 °C/min. The O.D. at 600 nm was recorded every 5 °C...”

Supplementary Fig. 5. The effects of temperature on the turbidity of JSF1 solutions. (a) Turbidity of 0.5–5 mM JSF1 at 5–40 °C. (b) The temperature-dependent reversibility of 3 mM JSF1 coacervates.

Comment 2: A 405nm laser, close to the photoinitiation wavelength of 365nm, was used in the FLIP experiments. Meanwhile, according to the absorption data in Figure S6,

JSF1 has a non-zero absorbance at around 400nm. In this case, it would be better to provide some evidence showing the impact of the bleaching lasers on the cleavage of JSF1.

Response: We appreciated so much for this comment. It is true that 405 nm laser could also impact on the cleavage of JSF1. In order to avoid the possible impact from 405 nm laser, we have now added a new experiment by replacing the original FUS-488 solution with the mixture of FUS and FUS-488 (FUS-488:FUS = 1:9). With this new experimental condition, 480 nm laser was sufficient to bleach FUS-488 and 405 nm laser was no longer needed. In our new result, we also confirmed the high fluidity of FUS-488 and *f*-JSF1 in the mixture, suggesting that JSF1 underwent co-phase separation with FUS (**Fig. 3d and 3e**). The new experimental details could be found in **page 16, line 358-370**.

Revised Figure 3d and 3e

Comment 3: In the discussion part, it was mentioned that the LCD fragments and poly-R contributed to the interactions that "promote LLPS." However, the results showed increased fluidity and a change in number density. These influences are not precisely equal to the promotion of LLPS. Thus, the expression here might need some adjustment, or more evidence, such as the change of the critical temperature, salt concentration, or FUS concentration under the impact of JSF1, needs to be provided.

Response: We appreciated for this comment and have now added a few experiments to suggest JSF1 could promote the phase separation of FUS. While the critical concentration required for FUS LLPS is around 1.25 μM in the absence of JSF1 (**Fig. 3a**), the critical FUS concentration was reduced to 0.625 μM in the presence of JSF1 (detail in **Supplementary Fig. 19**).

In the revised manuscript (**page 14, line 275-277**), we have now replaced “We surmised that the FUS LCD ...significantly contribute to these multivalent interactions that promote LLPS” by “We surmised that the FUS LCD ...significantly contribute to these multivalent interactions which are essential for LLPS.”

Supplementary Fig. 19. JSF1 and RRRRRR lowered the critical concentration of FUS LLPS. (a) DIC images of FUS solution (0.625 μM), with JSF1 (25 μM), and with RRRRRR (25 μM). (b) the condensate density of FUS. (c) DIC images of JSF1 solution (25 μM) and RRRRRR solution (25 μM) without FUS.

Comment 4: The cell-based assays were performed with FUS-transfected cells only. Without the controls with the non-transfected conditions, it would be hard to determine the effect of the JSF1 treatment alone and the effect of light cleavage alone to the cell toxicity. Thus, it's essential to include the controls with -FUS-eGFP, + JSF1 and +/- light cleavage.

Response: We appreciated for this suggestion and have now added new experiments to identify the cell toxicity of all the controls under the non-transfected conditions (**Supplementary Fig. 23b**). As shown in these results, all controls (with -FUS-eGFP, +/- JSF1, and +/- light cleavage) exhibited negligible toxicity.

Supplementary Fig. 23. The cell toxicity of N2A cells (a) treated with different concentration (0–20 μM) of JSF1 and (b) cells in the presence or absence of JSF1 (10μM) with or without photoinitiation. The statistic results were shown as mean ± SD

(n = 4). Data were analyzed by one-way ANOVA with Tukey post-hoc test. n.s.: non-significant.

In the revised manuscript (**page 12, line 234-235**), we have now added: “Meanwhile, all control experiments with the non-transfected conditions exhibited negligible toxicity (Supplementary Fig. 23b).”

Related References:

- Babinchak, W. M., Dumm, B. K., Venus, S., Boyko, S., Putnam, A. A., Jankowsky, E., & Surewicz, W. K. (2020). Small molecules as potent biphasic modulators of protein liquid-liquid phase separation. *Nature Communications*, *11*(1), 5574. <https://doi.org/10.1038/s41467-020-19211-z>
- Babinchak, W. M., Haider, R., Dumm, B. K., Sarkar, P., Surewicz, K., Choi, J. K., & Surewicz, W. K. (2019). The role of liquid-liquid phase separation in aggregation of the TDP-43 low-complexity domain. *J Biol Chem*, *294*(16), 6306-6317. <https://doi.org/10.1074/jbc.RA118.007222>
- Gu, J., Liu, Z., Zhang, S., Li, Y., Xia, W., Wang, C., Xiang, H., Liu, Z., Tan, L., Fang, Y., Liu, C., & Li, D. (2020). Hsp40 proteins phase separate to chaperone the assembly and maintenance of membraneless organelles. *Proc Natl Acad Sci U S A*, *117*(49), 31123-31133. <https://doi.org/10.1073/pnas.2002437117>
- Li, Y., Gu, J., Wang, C., Hu, J., Zhang, S., Liu, C., Zhang, S., Fang, Y., & Li, D. (2022). Hsp70 exhibits a liquid-liquid phase separation ability and chaperones condensed FUS against amyloid aggregation. *iScience*, *25*(6), 104356. <https://doi.org/10.1016/j.isci.2022.104356>
- Liu, Z., Zhang, S., Gu, J., Tong, Y., Li, Y., Gui, X., Long, H., Wang, C., Zhao, C., Lu, J., He, L., Li, Y., Liu, Z., Li, D., & Liu, C. (2020). Hsp27 chaperones FUS phase separation under the modulation of stress-induced phosphorylation. *Nat Struct Mol Biol*, *27*(4), 363-372. <https://doi.org/10.1038/s41594-020-0399-3>
- Mary Wilcox, R. W. V., Katherine W. Johnson, Andrew P. Billington, Barry K. Carpenter, James A. McCray, Anthony P. Guzikowski, and George P. Hess. (1990). Synthesis of photolabile precursors of amino acid neurotransmitters. *The Journal of Organic Chemistry*, *55*(5), 1585-1589. <https://doi.org/https://doi.org/10.1021/jo00292a038>
- Michieletto, D., & Marena, M. (2022). Rheology and Viscoelasticity of Proteins and Nucleic Acids Condensates. *JACS Au*, *2*(7), 1506-1521. <https://doi.org/10.1021/jacsau.2c00055>
- Mitrea, D. M., Mittasch, M., Gomes, B. F., Klein, I. A., & Murcko, M. A. (2022). Modulating biomolecular condensates: a novel approach to drug discovery. *Nat Rev Drug Discov*, *21*(11), 841-862. <https://doi.org/10.1038/s41573-022-00505-4>
- Murakami, T., Qamar, S., Lin, J. Q., Schierle, G. S., Rees, E., Miyashita, A., Costa, A. R., Dodd, R. B., Chan, F. T., Michel, C. H., Kronenberg-Versteeg, D., Li, Y., Yang, S. P., Wakutani, Y., Meadows, W., Ferry, R. R., Dong, L., Tartaglia, G. G., Favrin, G., . . . St George-Hyslop, P. (2015). ALS/FTD Mutation-Induced

Phase Transition of FUS Liquid Droplets and Reversible Hydrogels into Irreversible Hydrogels Impairs RNP Granule Function. *Neuron*, 88(4), 678-690. <https://doi.org/10.1016/j.neuron.2015.10.030>

Schuster, B. S., Reed, E. H., Parthasarathy, R., Jahnke, C. N., Caldwell, R. M., Bermudez, J. G., Ramage, H., Good, M. C., & Hammer, D. A. (2018). Controllable protein phase separation and modular recruitment to form responsive membraneless organelles. *Nature Communications*, 9(1), 2985. <https://doi.org/10.1038/s41467-018-05403-1>

Shen, Y., Chen, A., Wang, W., Shen, Y., Ruggeri, F. S., Aime, S., Wang, Z., Qamar, S., Espinosa, J. R., Garaizar, A., St George-Hyslop, P., Colleparado-Guevara, R., Weitz, D. A., Vigolo, D., & Knowles, T. P. J. (2023). The liquid-to-solid transition of FUS is promoted by the condensate surface. *Proc Natl Acad Sci U S A*, 120(33), e2301366120. <https://doi.org/10.1073/pnas.2301366120>

Reviewers' Comments:

Reviewer #1:

Remarks to the Author:

In their revised manuscript, the authors have addressed my concerns. I support publication in Nature Communications.

Reviewer #2:

Remarks to the Author:

The revised manuscript has addressed some of the original critiques raised by this reviewer. However, there are several issues and the quantitative nature of the studies dampens my enthusiasm about this manuscript. Specifics are provided below:

The design principle of the peptide is still questionable, are there a solid rationale for the design of the peptide? It seems like the authors used some of the verbiage from my review to explain their narrative in a few places.

Comment 3: It is still unclear what is Arg6 doing to FUS condensate liquid to fiber formation, If it lowers C_{sat} of FUS it would mean that FUS is now quenched deeper in a two-phase regime, which will change the kinetics of condensate maturation. Does cleaving the Arg6 tag at different time points matter?

Comment # 5. The FUS50-60 peptide forms fiber as their new data shows, could it be templating the FUS fiber formation? If yes, then how? Are these heterotypic FUS-peptide fibers? Without a structural model, these findings limit our understanding of the system.

Comment 7: Many questions remain about what is happening in cells due to the highly qualitative nature of these data and their analysis. For example, it is still unclear why the FUS cellular condensates are more dynamic. Why FUS condensates are less toxic in the presence of the peptide? The link between FUS aggregation and toxicity is very vague as portrayed.

Reviewer #3:

Remarks to the Author:

The authors have satisfactorily addressed all of the concerns raised by the reviewer.

Response to Reviewer 1:

In their revised manuscript, the authors have addressed my concerns. I support publication in Nature Communications.

Response:

We appreciate for all the suggestions and comments from Reviewer 1, which have significantly improved our manuscript.

Response to Reviewer 2:

Comment 1: The design principle of the peptide is still questionable, are there a solid rationale for the design of the peptide? It seems like the authors used some of the verbiage from my review to explain their narrative in a few places.

Response:

The design principle of the peptide is based on (a) the intrinsic nature of FUS protein (b) the multivalent interactions for coacervation from literatures. To further evaluate the suitable sequences from FUS protein, we have also applied (c) various bioinformatics tools (e.g., PONDR, PLAAC, PScore, and PASTA 2.0) in this work.

The details are shown as the followings:

“Among different DNA/RNA-binding proteins, FUS stands out due to its rapid recruitment to DNA damage sites, its ability to form droplets both in vivo and in vitro, and its propensity for liquid-to-solid transitions, ultimately resulting in disease-related aggregates under pathological conditions. Studies have emphasized the significance of tyrosine residues (denoted Y) in LCDs and arginine residues (denoted R) in arginine–glycine–glycine-rich (RGG) domains (Fig. 1a) in dominating its LLPS properties through cation– π and π – π interactions^{8, 19}. Furthermore, the antiparallel β -sheet motif in the LCD (FUS_{39–95}) was proven to play a key role in FUS self-assembly. Additionally, short fragments of the FUS LCD (FUS_{54–59} and FUS_{50–65}) that can form either reversible or irreversible aggregates have been observed.

The phase modulator, JSF1, was mainly composed of a FUS LCD fragment and a polyarginine tract (RRRRRR). By applying different protein aggregation predictors, we identified the protein segments (FUS_{50–60}: YGQSSYSSYGQ) in FUS LCD with high propensity for fibrilization (Supplementary Fig. 1). We surmised the multivalent interactions (e.g., cation– π and π – π interactions) between positively charged arginines and the three tyrosines in FUS_{50–60} could benefit greatly on the droplet formation. Meanwhile, the polyarginine tract could also provide cell penetrating ability. To further enable the photocontrollable ability of this phase modulator, the photocleavable linker was applied to conjugate FUS LCD fragment with the polyarginine tract.

Recent studies have elucidated the driving force behind FUS condensation. Through the sequence analysis, truncation studies, and FUS mutation analysis, the LLPS of FUS

has been shown to be primarily driven by the LCD and RGG domains of FUS (Fig. 1a) with cation- π and π - π interactions. We surmised that the FUS LCD fragment (FUS₅₀₋₆₀) and the cationic polyarginine tract significantly contribute to these multivalent interactions which are essential for LLPS. In addition, through the use of various bioinformatics tools (e.g., PONDR, PLAAC, PScore, and PASTA 2.0), FUS₅₀₋₆₀ was a highly disordered sequence (Supplementary Fig. 27a) that has prion-like properties (Supplementary Fig. 27b), strong π - π interactions (Supplementary Fig. 27c), and a high tendency to form aggregate (Supplementary Fig. 1). These features likely account for the amyloid-like properties (Fig. 2) and seeding capacity (Fig. 3f-3h) of the photocleaved FUS₅₀₋₆₀ peptide from JSF1.”

In addition, we have removed all the specific wording from Reviewer 2’s previous comments in the revised manuscript.

Comment 2: It is still unclear what is Arg6 doing to FUS condensate liquid to fiber formation, If it lowers C_sat of FUS< it would mean that FUS is now quenched deeper in a two-phase regime, which will change the kinetics of condensate maturation. Does cleaving the Arg6 tag at different time points matter?

Response:

The role of Arg6 in FUS condensate maturation is complicated. Although the structure of FUS LCD fibril at a thermodynamically stable state has been proposed¹, the kinetic details of FUS condensate maturation has remained unclear^{2,3}. Currently, it is still hard to characterize the influence of Arg6 on the kinetics of condensation maturation. This is also why we focus on the fibril formation of FUS after sufficient incubation upon the regulation of JSF1 after 48 h (**Fig. 3f**). In addition, even in the presence of FUS fibrils, the ThT fluorescence signal is still too weak for kinetic studies. However, since JSF1 cannot trigger the fibrilization of FUS without photolysis, we surmise the release of Arg6 at a later time point may postpone the formation of FUS fibrils.

Comment 3. The FUS50-60 peptide forms fiber as their new data shows, could it be templating the FUS fiber formation? If yes, then how? Are these heterotypic FUS-peptide fibers? Without a structural model, these findings limit our understanding of the system.

Response: Since the fibrils were generated by the seeding effects of photolyzed JSF1, we believe that the FUS₅₀₋₆₀ peptide could template the formation of heterotypic FUS-peptide fibrils. Since the main focus of this manuscript is methodology development

and we do not have instant access of CryoEM, we will not be able to provide any structural model in this manuscript.

Comment 4: Many questions remain about what is happening in cells due to the highly qualitative nature of these data and their analysis. For example, it is still unclear why the FUS cellular condensates are more dynamic. Why FUS condensates are less toxic in the presence of the peptide? The link between FUS aggregation and toxicity is very vague as portrayed.

Response: Based on the reported literature, **the increase of FUS condensates immobile fraction** is correlated with cell viability and FUS proteinopathy⁴⁻⁶. In short, the maturation of condensates will result in higher immobile fraction and lower cell viability. In fact, we found mutant FUS condensates, which contained 40% immobile fraction (Fig. 4e) are toxic (viability = 70%) to the cells (Fig. 4f). The addition of JSF1 will reduce the immobile fraction to 20%, which increase the viability to 80%. By contrast, in the presence of photoinitiated JSF1 increase the immobile fraction to 60% which decrease the viability to 60%, consisting with the reported literatures.

We surmise that JSF1, due to its high valency and additional crosslinking between FUS scaffolds, increase the mobility of FUS condensates, potentially altered the toxic events caused by matured FUS condensates. Nevertheless, the mechanisms of FUS condensation and maturation related to toxicity are still not fully understood. Proposed mechanisms involve interactions with RNAs, chaperones, and other DNA/RNA binding proteins, but these remain controversial. The involvement of these molecules during the condensation and maturation process will be explored in the future. Currently, we are sharing this new tool to accelerate the related studies.

(1) Murray, D. T.; Kato, M.; Lin, Y.; Thurber, K. R.; Hung, I.; McKnight, S. L.; Tycko, R. Structure of FUS Protein Fibrils and Its Relevance to Self-Assembly and Phase Separation of Low-Complexity Domains. *Cell* **2017**, *171* (3), 615-627 e616. DOI: 10.1016/j.cell.2017.08.048.

(2) Chatterjee, S.; Kan, Y.; Brzezinski, M.; Koynov, K.; Regy, R. M.; Murthy, A. C.; Burke, K. A.; Michels, J. J.; Mittal, J.; Fawzi, N. L.; et al. Reversible Kinetic Trapping of FUS Biomolecular Condensates. *Adv Sci (Weinh)* **2022**, *9* (4), e2104247. DOI: 10.1002/advs.202104247 From NLM Medline.

- (3) Berkeley, R. F.; Kashefi, M.; Debelouchina, G. T. Real-time observation of structure and dynamics during the liquid-to-solid transition of FUS LC. *Biophys J* **2021**, *120* (7), 1276-1287. DOI: 10.1016/j.bpj.2021.02.008 From NLM Medline.
- (4) Zhang, P.; Fan, B.; Yang, P.; Temirov, J.; Messing, J.; Kim, H. J.; Taylor, J. P. Chronic optogenetic induction of stress granules is cytotoxic and reveals the evolution of ALS-FTD pathology. *Elife* **2019**, *8*. DOI: 10.7554/eLife.39578 From NLM Medline.
- (5) Murakami, T.; Qamar, S.; Lin, J. Q.; Schierle, G. S.; Rees, E.; Miyashita, A.; Costa, A. R.; Dodd, R. B.; Chan, F. T.; Michel, C. H.; et al. ALS/FTD Mutation-Induced Phase Transition of FUS Liquid Droplets and Reversible Hydrogels into Irreversible Hydrogels Impairs RNP Granule Function. *Neuron* **2015**, *88* (4), 678-690. DOI: 10.1016/j.neuron.2015.10.030.
- (6) Molliex, A.; Temirov, J.; Lee, J.; Coughlin, M.; Kanagaraj, A. P.; Kim, H. J.; Mittag, T.; Taylor, J. P. Phase Separation by Low Complexity Domains Promotes Stress Granule Assembly and Drives Pathological Fibrillization. *Cell* **2015**, *163* (1), 123-133. DOI: 10.1016/j.cell.2015.09.015.

Response to Reviewer 3:

The authors have satisfactorily addressed all of the concerns raised by the reviewer.

Response:

We appreciate for all the constructive comments from Reviewer 3, which have significantly enhanced the quality of our manuscript.